# Serine metabolism remodeling after platinum-based chemotherapy identifies vulnerabilities in a subgroup of resistant ovarian cancers

Tom Van Nyen [1,2], Mélanie Planque[3,4], Lilian van Wagensveld [5,6,7], Joao A. G. Duarte[3,4], Esther A. Zaal [8,9], Ali Talebi[10], Matteo Rossi[3,4], Pierre-René Körner[2], Lara Rizzotto[11], Stijn Moens[1], Wout De Wispelaere[1], Regina E. M. Baiden-Amissah [1], Gabe S. Sonke[12], Hugo M. Horlings [13], Guy Eelen [14], Emanuele Berardi[15], Johannes V. Swinnen [10], Celia R. Berkers [8,9], Peter Carmeliet [14,16,17], Diether Lambrechts [18,19], Ben Davidson[20,21], Reuven Agami [2,22], Sarah-Maria Fendt [3,4], Daniela Annibali [1,2,25] ✉ & Frédéric Amant [1,23,24,25] ✉

Resistance to platinum-based chemotherapy represents a major clinical challenge for many tumors, including epithelial ovarian cancer. Patients often experience several response-relapse events, until tumors become resistant and life expectancy drops to 12–15 months. Despite improved knowledge of the molecular determinants of platinum resistance, the lack of clinical applicability limits exploitation of many potential targets, leaving patients with limited options. Serine biosynthesis has been linked to cancer growth and poor prognosis in various cancer types, however its role in platinum-resistant ovarian cancer is not known. Here, we show that a subgroup of resistant tumors decreases phosphoglycerate dehydrogenase (PHGDH) expression at relapse after platinum-based chemotherapy. Mechanistically, we observe that this phenomenon is accompanied by a specific oxidized nicotinamide adenine dinucleotide ($NAD^+$) regenerating phenotype, which helps tumor cells in sustaining Poly (ADP-ribose) polymerase (PARP) activity under platinum treatment. Our findings reveal metabolic vulnerabilities with clinical implications for a subset of platinum resistant ovarian cancers.

Resistance to conventional chemotherapy is an obstacle for the treatment of many cancer types[1]. Epithelial ovarian cancer, the second most lethal gynecological malignancy worldwide[2], is a paradigmatic example. For more than four decades, standard of care first-line treatment is debulking surgery and platinum-based chemotherapy, and 5 years-overall survival rates remain close to 45%. High-grade serous carcinoma (HGSC) accounts for 70% of all ovarian cancer cases and most of the deaths. Although initial response to chemotherapy is very high (~80%), eventually patients relapse and succumb to platinum-resistant disease[3]. Over the years, important steps have led to a better understanding of the biology underlying resistance to platinum, but they have had only modest clinical impact and for resistant patients treatment objectives remain disease control and palliation[4–7].

Metabolic alterations (e.g., Warburg effect, addiction to glutamine, or increased serine biosynthesis) have been linked to tumor development and maintenance in different cancer types[8–12]. Since metabolism is intertwined to signaling pathways controlling cell death[13], it is becoming more evident that metabolic adaptations may also support resistance to both chemotherapeutics and targeted agents[14–16]. Multiple and different traits, involving adaptations in both glucose and glutamine metabolism, and mitochondrial activity have been associated with platinum resistance in ovarian cancer cells[17–19]. Nevertheless, a concrete plan for their clinical translatability is still missing. Recent reports highlighted the role that reprogramming of the metabolism of certain amino acids might play in tumor adaptation to environmental cues and eventually in anticancer drug response[20]. The biosynthesis of serine from glucose, mediated by phosphoglycerate dehydrogenase (PHGDH), is one of the most studied metabolic pathways and it has been linked to development and progression of different cancer types[12,21–24]. Of note, it has been shown that primary and metastatic breast cancers differ in their serine biosynthetic requirements for molecular pathways activation, and that this may influence their sensitivity to specific targeted therapies[25], while heterogeneous and lower PHGDH expression potentiates tumor dissemination and metastasis formation in breast cancer[26]. However, the relevance of PHGDH expression and serine biosynthesis activity in tumors that acquire resistance after platinum exposure is largely unexplored.

Analyzing matched samples collected longitudinally during disease course, we found that a subgroup of ovarian cancer patients relapsed after platinum-based chemotherapy are characterized by decreased intratumor PHGDH levels, and that the extent of such decrease correlates with worse prognosis. We observed in in vitro preclinical models that PHGDH downregulation is accompanied by a general switch in the central carbon metabolism regulating NAD$^+$ availability, and that this may help to sustain PARP activity under platinum-based treatment. Consequently, combining carboplatin and PARP or NAD$^+$ synthesis inhibitors affects the growth of resistant models with decreased serine synthesis activity. Together, our results identify alterations in NAD$^+$ and serine metabolism as actionable vulnerabilities of a fraction of ovarian tumors adapting after platinum exposure, and provide a rationale to test new therapeutic approaches to overcome resistance.

## Results

### Decrease in PHGDH expression at relapse after platinum-based chemotherapy identifies a subset of ovarian cancer patients

To investigate serine biosynthetic activity in platinum-resistant ovarian cancers, we assessed gene expression levels of *PHGDH* (mRNA) in the original HGSC dataset from TCGA[27], with $n = 287$ newly diagnosed cases reporting platinum status. These data showed heterogeneity in PHGDH levels in chemo-naïve tumors, and that surprisingly patients who relapsed within 6 months since their last platinum treatment (clinically resistant cases, platinum-free interval PFI < 6 months, $n = 90$) had significantly lower *PHGDH* mRNA expression than the sensitive ones (PFI > 6 months, $n = 197$) (Fig. 1a). To investigate to what extent PHGDH expression might change during the process of developing resistance to platinum, we assessed PHGDH protein levels by immunohistochemistry (IHC) on pairs of archival effusion specimens sampled in the primary setting (chemo-naïve) and at the moment they were declared resistant, from nine distinct HGSC patients who received standard of care platinum-based chemotherapy as first-line treatment at the University of Oslo, Norway (Supplementary Table 1). Scoring from an expert pathologist showed that 6 out of 9 patients had decreased PHGDH levels at recurrence, while in 2 patients PHGDH scores did not change and only 1 had increased PHGDH at recurrence (Fig. 1b, c and Supplementary Fig. 1a, b and Supplementary Table 1). Although all patients died of disease, the four patients with the most pronounced decrease in PHGDH (pair 3, 5, 6 and 7) had the lowest

overall survival (Fig. 1d and Supplementary Table 1). These data show that PHGDH downregulation may occur during disease progression in a subgroup of platinum-resistant ovarian tumors, and that the extent of decrease correlated with worse survival. We then sought to validate our findings in a second, larger cohort of patients from The Netherlands Cancer Institute (Amsterdam, The Netherlands). Since currently a large number of newly-diagnosed patients receive neoadjuvant platinum-based therapy, this new and more heterogeneous cohort consisted of 25 paired tumor tissue biopsies, collected at diagnosis (primary debulking or after neoadjuvant platinum treatment), and at relapse (early or late) (Fig. 1b and Supplementary Table 2). In this Dutch cohort, scored independently by a second expert pathologist, we also observed that PHGDH downregulation at recurrence was an appreciable phenomenon (Fig. 1b, e and Supplementary Fig. 1c, d and Supplementary Table 2), with PHGDH scores decreased in 11 patients, increased in 12 patients, and unchanged in 2 patients. In addition, patients with decreased PHGDH scores at recurrence had lower overall survival, compared to patients who had increased PHGDH scores (Supplementary Fig. 1e and Supplementary Table 2) and we confirmed the correlation between the extent of PHGDH levels decrease and overall survival (Fig. 1f). Together, we found that PHGDH downregulation is an event happening in ~44–67% of the patients who received platinum-based chemotherapy in two independent cohorts of longitudinally-collected paired biopsies. This suggests that decreased PHGDH expression at recurrence after platinum exposure, independent of primary PHGDH levels, could be associated to development of resistance.

### Platinum-resistant ovarian cancer cells characterized by decreased serine biosynthetic activity are auxotrophic for serine

We then hypothesized that platinum exposure could remodel serine metabolism. To further investigate serine metabolism in acquired platinum resistance while mimicking the setting of our clinical paired datasets, we used the A2780wt cell line and its isogenic resistant counterpart A2780cis, originally established by chronically exposing parental cells to cisplatin[28], and tested their response to carboplatin. Resistant cells grew slower than parental cells but were able to sustain proliferation under acute carboplatin treatment, reflected in their GI50 values ($17.09 \pm 1.11 \, \mu M$ for cis and $2.85 \pm 1.11 \, \mu M$ for wt cells, respectively) (Supplementary Fig. 2a). Next, we analyzed the expression of the three serine biosynthetic enzymes and found that PHGDH and phosphoserine aminotransferase (PSAT1) were downregulated in resistant cells compared to sensitive cells both at the protein (Fig. 2a) and mRNA level (Fig. 2b), while phosphoserine phosphatase (PSPH) expression was decreased only at the protein level (Fig. 2a, b), suggesting a post-transcriptional regulation for this specific enzyme. To investigate the metabolic changes associated with resistance and low PHGDH expression, we supplemented both cell lines with $^{13}C_6$-glucose and extracted metabolites at day 5 of carboplatin treatment (Supplementary Fig. 2b). Abundance of specific metabolites (i.e., glycolytic and TCA intermediates, amino acids and lipids) was altered in sensitive cells under treatment but also in untreated resistant cells, suggesting that sensitive cells undergo metabolic changes after platin exposure, and that specific rearrangements could be associated with acquired resistance (Supplementary Fig. 2c). Specifically, we detected striking alterations in the isotopomer distributions of serine and glycine derived from $^{13}C_6$-glucose (Fig. 2c–e), with resistant cells showing almost no incorporation of $^{13}C$ in any of the two amino acids compared to sensitive cells (Fig. 2d, e), indicating that the process of de novo serine (and glycine) synthesis from glycolytic intermediates was not active in A2780 cells once they acquired resistance (Supplementary Fig. 2d). In resistant cells, downregulation of serine biosynthesis correlated with lower intracellular abundance of both serine and glycine (Fig. 2f, g). In addition, we observed increased glucose uptake (Fig. 2h) in resistant cells, suggesting that the decreased glucose-derived

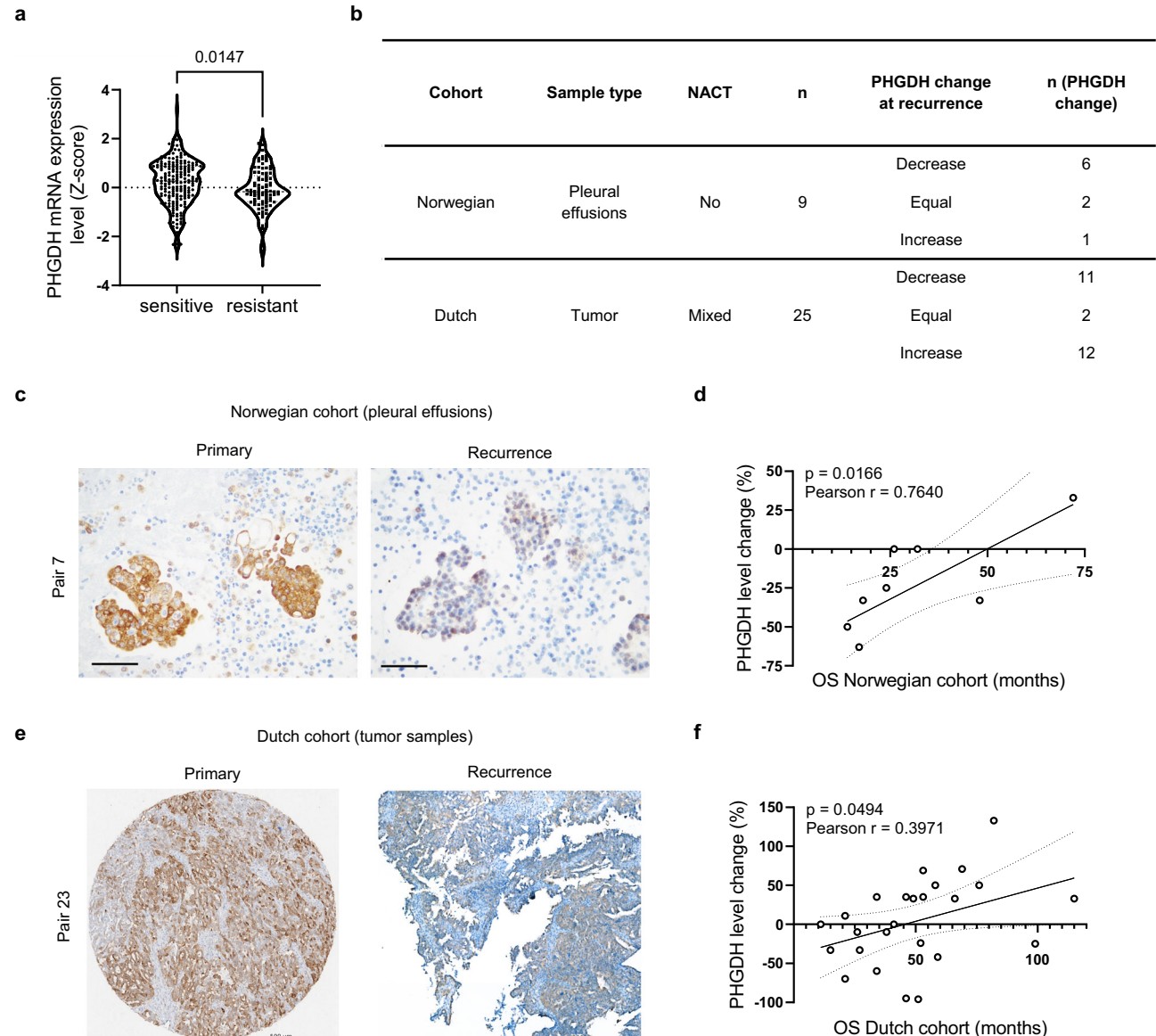

**Fig. 1 | Decrease in PHGDH expression after platinum-based chemotherapy identifies a subset of ovarian cancer recurrences. a** PHGDH mRNA expression levels of ovarian cancer patients in the TCGA cohort, $n = 197$ sensitive patients and $n = 90$ resistant patients, unpaired two-tailed $t$-test. **b** Overview of the Norwegian and Dutch patient cohort and related PHGDH changes, NACT neoadjuvant chemotherapy. **c** Example of PHGDH IHC staining of primary and late recurrent clinical specimen of the Norwegian cohort, scale bar is 50 µm. **d** Correlation of PHGDH protein level change at recurrence and overall survival in the Norwegian cohort, simple linear regression determined by minimizing sum of squares is shown with 95% confidence bands, Pearson correlation $r = 0.7640$ and $p = 0.0166$. **e** Example of PHGDH IHC staining of primary and recurrent clinical specimen of the Dutch cohort, scale bar is 500 µm. **f** Correlation of PHGDH protein level change at recurrence and overall survival in the Dutch cohort, simple linear regression determined by minimizing sum of squares is shown with 95% confidence bands, Pearson correlation $r = 0.3971$ and $p = 0.0494$. Source data are provided as a Source Data file. NACT neoadjuvant chemotherapy, OS overall survival.

carbons flux through the serine synthesis pathway (SSP) was not the consequence of lower intracellular glucose availability, but a peculiar metabolic adaptation. We also observed that resistant cells took up twice more serine from the medium, compared to the sensitive ones (Fig. 2i), as expected due to the higher expression of two major serine transporters, *SLC1A4* and *SLC1A5* (Fig. 2b), while this was not the case for other amino acids or for glycine, which was on the contrary excreted in the medium (Supplementary Fig. 2e). Of note, metabolic analysis at an early time point, after 24 h, showed the same trends observed analyzing $^{13}$C labeling at steady-state after 5 days (Supplementary Fig. 2f–h). Together, these data show that the acquisition of resistance to platinum in A2780 cells is associated with de novo serine synthesis downregulation and with an adaptive increase in extracellular serine uptake.

## Resistant A2780 cells require exogenous serine for nucleotides production and proliferation

Since our data showed that A2780cis cells do not synthetize serine, we then asked whether serine/glycine starvation could impair their growth and viability and observed that resistant cells deprived of serine and glycine did not proliferate (Fig. 2j), while sensitive cells continued to grow (Supplementary Fig. 3a). Supplementation of 0.093 mM serine, a fraction equivalent to 1/3 of the amount originally present in conventional medium (0.28 mM), was enough to fully recover the proliferative phenotype, while glycine supplementation was not, in line with what has been reported by Labuschagne et al. for different cells deprived from serine (Fig. 2j)[29]. Carboplatin treatment synergized with serine starvation, leading to the death of resistant cells, a phenotype that was again completely rescued by re-

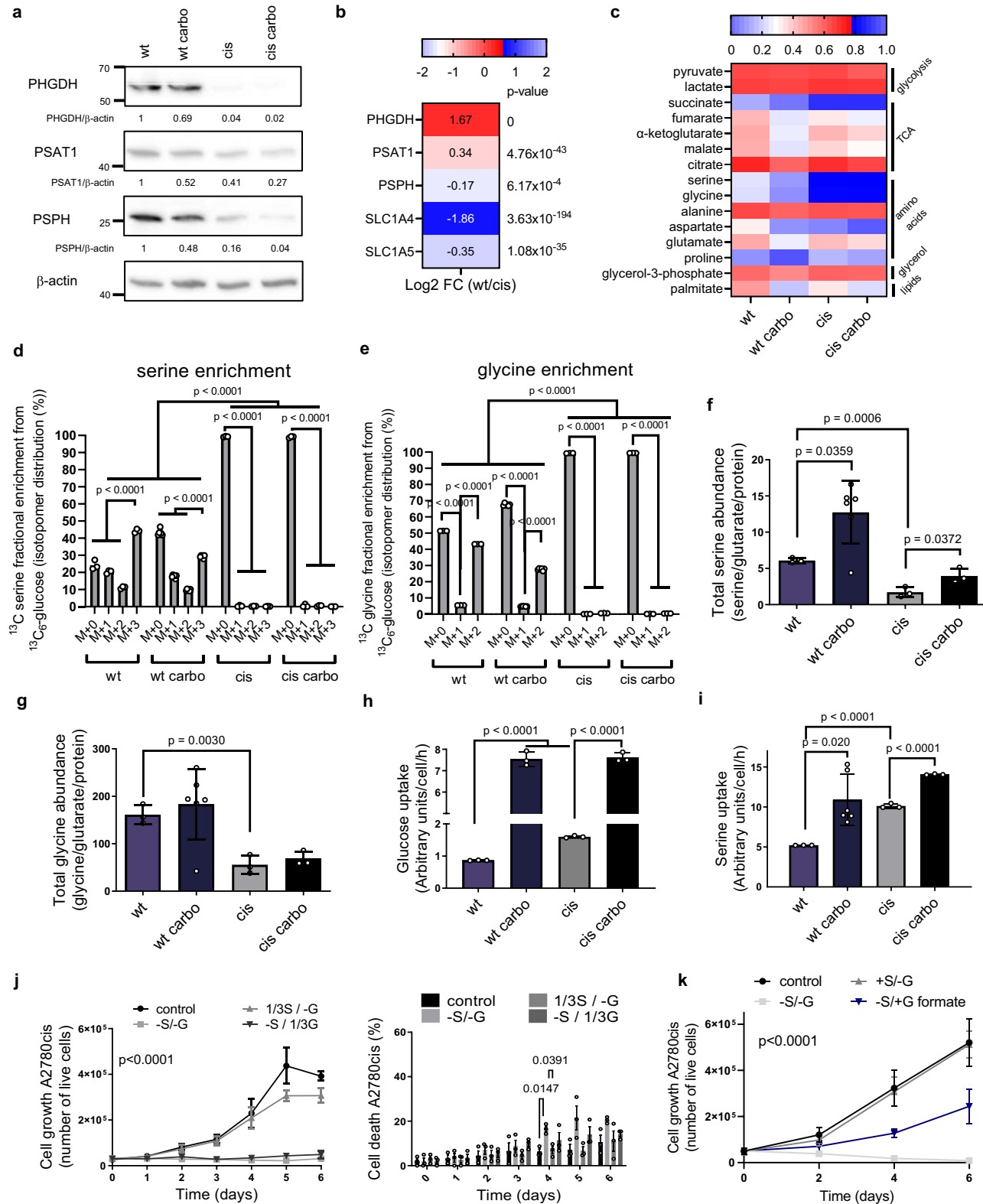

supplementing serine (Supplementary Fig. 3b). Thus, we concluded that resistant cells require extracellular serine to proliferate and survive platinum treatment.

Next, we asked whether serine/glycine deprivation selectively inhibited the growth of resistant cells by affecting their redox homeostasis, because we observed that platinum treatment induced oxidative stress in sensitive cells while resistant cells were able to better tolerate platinum-induced reactive oxygen species (ROS)

(Supplementary Fig. 3c, d). In resistant cells we observed increased levels of both cytoplasmic and mitochondrial ROS, but only under serine/glycine starvation (Supplementary Fig. 3c, d). In line, carboplatin treatment synergized with serine/glycine deprivation to induce ROS accumulation in both sensitive and resistant cells at day 5 (Supplementary Fig. 3c, d). However, although N-acetyl-cysteine (NAC) supplementation was able to rescue survival and proliferation of the sensitive cells under carboplatin (Supplementary Fig. 3e), it had no

**Fig. 2 | Ovarian cancer cells with acquired resistance after platinum exposure have low serine biosynthetic activity and are auxotrophic for serine.**
**a** Representative western blot of serine biosynthetic enzymes, $n = 5$ biological replicates. **b** Serine biosynthetic enzymes and main serine transporters enrichment in wt vs cis cells determined by RNA-seq analysis, mean values are represented, $n = 3$ technical replicates, $p$ values obtained by DESeq2 (Wald-test with Benjamini and Hochberg multiple testing). **c** Glucose enriched metabolites in wt and cis cells with and without 6 μM carboplatin determined by GC-MS, median values are represented, representative figure of $n = 3$ technical replicates and $n = 6$ technical replicates for wt carbo ($n = 2$ biological replicates). Serine (**d**) and glycine (**e**) isotopomer distribution in cells grown with $^{13}C_6$-labeled glucose, representative figure of $n = 3$ technical replicates and $n = 6$ technical replicates for wt carbo ($n = 2$ biological replicates), ordinary two-way ANOVA with Tukey's multiple comparison post-test, data are represented as mean ± SD, $p < 0.0001$. Total serine (**f**) and glycine (**g**) levels determined by GC-MS with glutarate as internal standard and normalized to protein level, representative figure of $n = 3$ ($n = 6$ for wt carbo) technical replicates ($n = 4$

biological replicates), unpaired two-tailed $t$-tests, data are represented as mean ± SD. **h** Glucose uptake determined by LC-MS, $n = 3$ technical replicates, unpaired two-tailed $t$-tests, data are represented as mean ± SD. **i** Serine uptake determined by GC-MS, $n = 3$ ($n = 6$ for wt carbo) technical replicates, unpaired two-tailed $t$-tests, data are represented as mean ± SD. **j** Growth (left) and cell death (right) of cis cells under serine/glycine deprivation, determined with trypan blue exclusion assay, $n = 3$ biological replicates (with three technical replicates each), repeated measures two-way ANOVA, $p < 0.0001$ (left) and repeated measures two-way ANOVA with Tukey's multiple comparison post-test, $p = 0.0393$ (right), data are represented as mean ± SEM. **k** Growth of cis cells in serine/glycine deprived medium supplemented with 1 mM formate determined by trypan blue counting, $n = 3$ biological replicates (with three technical replicates each), repeated measures two-way ANOVA, data are represented as mean ± SD, $p < 0.0001$. Source data are provided as a Source Data file. FC fold change, Carbo carboplatin treated, S serine, G glycine, TCA tricarboxylic acid cycle.

effect on resistant cells starved from serine/glycine (Supplementary Fig. 3f). These results suggest that in resistant cells the oxidative stress measured under serine/glycine deprivation, unlike the one caused by carboplatin treatment in sensitive cells, is not the primary cause of the proliferative arrest but arises as consequence of an earlier metabolic imbalance.

Then, we investigated to what extent the proliferative arrest of resistant cells grown without serine/glycine was caused by their decreased ability to produce nucleotides. Since serine is a major donor of one-carbon (1C) units and can be converted to glycine, both necessary for synthetizing purine precursors and thymine (Supplementary Fig. 3g, h), we supplemented resistant cells with formate and glycine, to define whether proliferation can be restored in the absence of serine. We observed a partial rescue of proliferation when serine-starved resistant cells were supplemented with both formate and glycine (Fig. 2k), suggesting that resistant cells require exogenous serine to produce nucleotides, in agreement to what has been previously reported in different cancer cells[29]. To verify this, we grew sensitive and resistant cells in the presence of $^{13}C_3$-serine and observed labeling incorporation in ~70% of the purines and TMP in resistant cells, and only ~15–20% in the sensitive ones (Supplementary Fig. 3i–l). Together, these data show that A2780 resistant cells rely on exogenous serine to produce nucleotides and sustain proliferation.

## Serine biosynthetic activity correlates with platinum responsiveness in A2780 resistant cells

Next, we investigated whether a longer starvation would result in a stronger effect, since 5 days of serine deprivation impaired proliferation specifically in resistant cells but did not induce cell death (Fig. 2j). After ±10 days in medium without serine/glycine, resistant cells adapted to bypass the growth arrest and started to proliferate again, when grown in the absence of carboplatin (Fig. 3a). These proliferating cells (named cis SSP cells) expressed serine biosynthetic enzymes, mainly PHGDH, at higher levels compared to the ones observed in resistant cells, both at protein and mRNA level (Fig. 3b, c). These data demonstrate that resistant cells could selectively switch their metabolism and re-activate serine biosynthesis from glucose in order to survive the amino acid deprivation, but only in the absence of platinum. As already reported for other cancer cell types under serine starvation[30,31], we observed that *PHGDH* re-expression was regulated at the transcriptional level following ATF4 upregulation at day 4, in serine/glycine-free medium (Fig. 3d). We then investigated whether serine biosynthesis re-activation would correlate with changes in platinum responsiveness. We found that cis SSP cells partially regained sensitivity to platinum when grown in serine/glycine depleted or complete medium, as shown by their GI50 value of 6.32 ± 1.04 μM and 8.42 ± 1.10 μM, respectively (Fig. 3e). These results show that resistant cells are able to re-activate serine synthesis, if subjected to the metabolic stress induced by serine

deprivation, and this partly re-sensitize them to platinum. To assess whether PHGDH expression levels in resistant cells would correlate with carboplatin sensitivity also in full medium and not only under metabolic stress, we genetically overexpressed PHGDH (Fig. 3f), and verified that PHGDH overexpression partially re-sensitized resistant cells to carboplatin (Fig. 3g). This was not the case when we overexpressed the catalytically-inactive form of the enzyme (catalytic-dead PHGDH (CD-PHGDH)) (Fig. 3f, g), indicating that PHGDH catalytic function was essential for carboplatin re-sensitization.

We then investigated whether PHGDH downregulation in chemo-naïve sensitive cells would per se make them more resistant to carboplatin. To do this, we silenced it by means of two distinct shRNAs (shPHGDH1 and shPHGDH2, with downregulation to ~50% and ~10% of the original levels, respectively). Independently on the degree of silencing, PHGDH downregulation not only did not induce resistance in sensitive cells, but slightly increased their sensitivity to carboplatin (Supplementary Fig. 4a, b). Interestingly, we also observed that PHGDH downregulation was sufficient to induce DNA damage (Supplementary Fig. 4c, d) even in the absence of platinum, in agreement with what was observed in PHDGH^high subsets of lung cancer[24]. These data confirmed that PHGDH catalytic activity may influences platinum responsiveness in cells that acquired resistance after treatment, but not in sensitive, chemo-naïve cells.

## Relapses after platinum-based chemotherapy exposure in patient-derived xenograft (PDX) models are characterized by decreased PHGDH expression

To validate our findings in vivo, we used clinically annotated PDXs established from ovarian cancer patients (Supplementary Fig. 5a and Supplementary Table 3). To test our hypothesis that exposure to platinum may induce a metabolic rewiring involving, among other processes, also downregulation of serine biosynthesis, we measured PHGDH levels in the OVC1a PDX model, established from a chemo-naïve platinum sensitive tumor, and in the matched OVC1b PDX model, established from the clinical relapse after platinum treatment in the same patient (Supplementary Table 3). In agreement with the observed clinical response, the OVC1b tumors in mice were still sensitive to platinum (Fig. 4a), but expressed lower levels of PHGDH (Fig. 4b), compared to OVC1a. Next, we tested the OVC3 PDX model, established from a patient that initially responded to platinum but relapsed within 6 months and was declared resistant (Supplementary Table 3). OVC3 initially respond to treatment, however the tumors in the mice relapsed between 6–15 weeks after the last platinum treatment, resembling the clinical response in the patient (Supplementary Fig. 5b). We analyzed PHGDH levels in vehicle-treated OVC3 mice and in OVC3 mice that underwent a chemo-relapse-chemo cycle, and we observed lower PHGDH levels in the relapsed samples, compared to the control ones (Fig. 4c and Supplementary Fig. 5b).

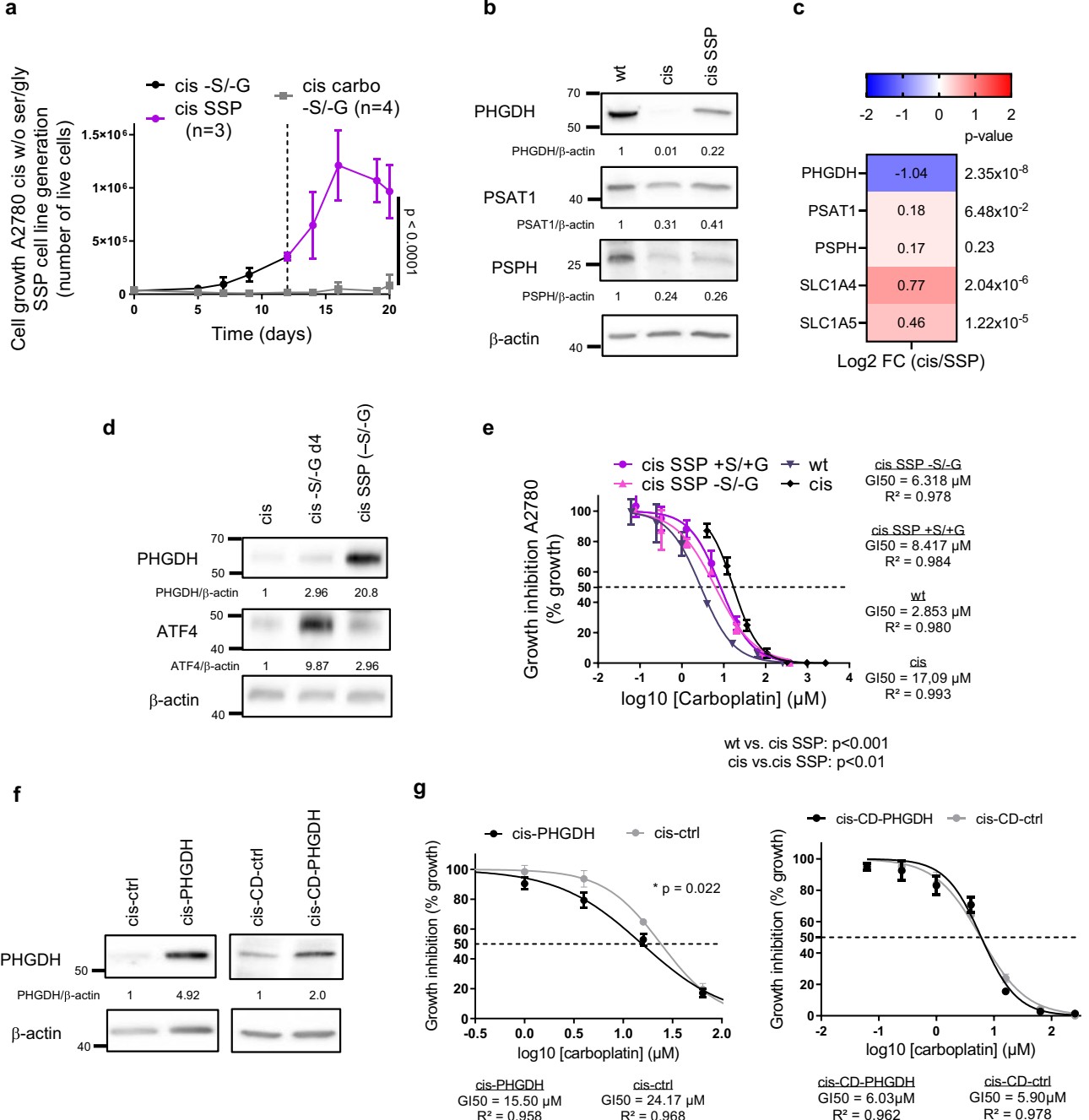

**Fig. 3 | Serine biosynthetic activity correlates with platinum responsiveness in A2780 resistant cells. a** Generation of cis SSP cell line established after serine/glycine deprivation of cis cells, $n = 3$ biological replicates for cis cells, $n = 4$ biological replicates for cis carbo cells (with three technical replicates each time), repeated measures two-way ANOVA, data are represented as mean ± SD. **b** Representative western blot of serine biosynthetic enzymes, $n = 4$ biological replicates. **c** RNA-seq results of serine biosynthetic enzymes and main transporters in cis SSP cells, mean values are plotted, $n = 3$ technical replicates, $p$ values obtained by DESeq2 (Wald-test with Benjamini and Hochberg multiple testing). **d** Representative western blot of PHGDH and ATF4 in cis cells, serine and glycine starved cis cells and SSP cells, $n = 2$ biological replicates. **e** GI50 curves of wt, cis and cis SSP cells, $n = 3$ biological replicates (with three technical replicates each time),

unpaired two-tailed $t$-tests between GI50 values, data are represented as mean ± SD, $p = 0.0047$ for cis vs. cis SSP +S/+G and $p = 0.0047$ for wt vs. cis SSP −S/−G. **f** Western blot of PHGDH and CD-PHGDH in genetic overexpressing cis cells, $n = 3$ biological replicates for cis-PHGDH and $n = 2$ biological replicates for cis-CD-PHGDH. **g** GI50 cuves of cis-control and cis-PHGDH overexpressing cells and cis-CD-control and cis-CD-PHGDH overexpressing cells, $n = 4$ biological replicates for cis-empty and cis-PHGDH and $n = 3$ biological replicates for cis-CD-empty and cis-CD-PHGDH (with three technical replicates each time), unpaired two-tailed $t$-tests between GI50 values, data are represented as mean ± SEM, $p = 0.022$ for cis-PHGDH vs cis-ctrl. Source data are provided as a Source Data file. Cis SSP serine synthesis pathway active cis cells, S serine, G glycine, CD catalytic dead.

## Platinum-resistant PDX tumors with low serine biosynthetic activity depend on exogenous serine for optimal growth

To investigate whether serine and glycine deprivation can affect in vivo the growth of tumors with low serine biosynthetic activity, we

compared two different PDX models, the sensitive OVC1a and the resistant OVC2 (PHGDH[low]) model, and exposed them to serine/glycine-free diet. OVC2 was established from a patient with a resistant lesion that developed during years, after being exposed to different

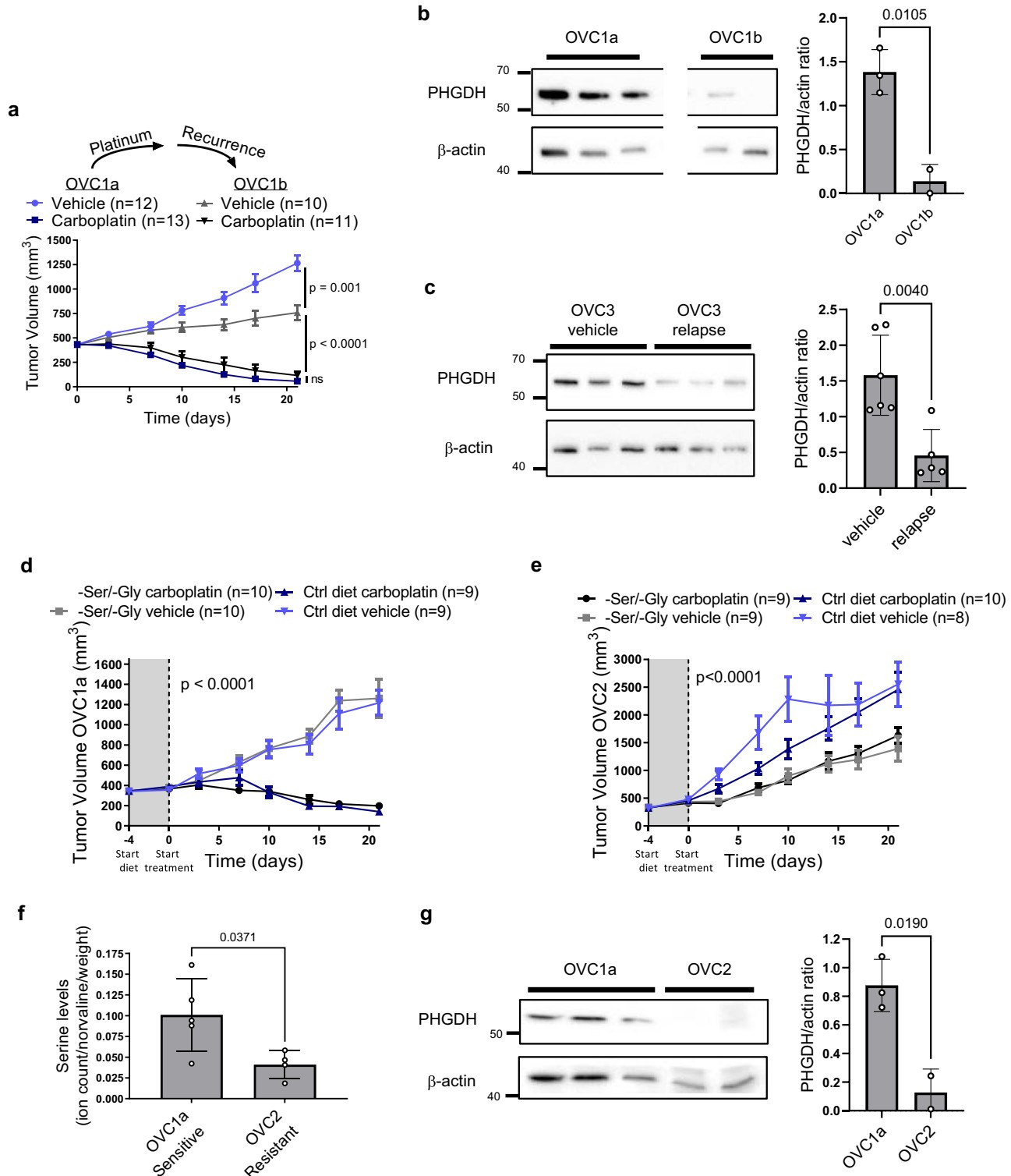

lines of platinum treatment (Supplementary Table 3). For both models we verified that response to platinum treatment effectively mimics the one observed in the original patient, with OVC1a tumors achieving complete response and OVC2 progressing under carboplatin treatment, respectively (Fig. 4d, e and Supplementary Table 3). We observed that resistant OVC2 tumors had significantly lower serine intracellular abundance and lower PHGDH levels, compared to the sensitive OVC1a ones (Fig. 4f, g). The serine/glycine-free diet increased the levels of all other amino acids in the plasma of our NMRI nude control mice, resulting in decreased relative serine and glycine levels,

but did not decrease the absolute serine and glycine concentration (Supplementary Fig. 5c, d). This could be explained by the fact that circadian fluctuations in circulating serine levels are at a low in the afternoon, when all our samples have been collected (4–6 p.m.) and that immunocompromised mice seem to have higher basal plasma levels of serine, compared to the C57BL/6J mice[22]. Although we did not see a synergistic effect with carboplatin administration, suggesting that different—and maybe longer—diet protocols could be needed, we observed that also in vivo resistant tumors grew significantly slower upon serine/glycine-free diet, while the sensitive ones were not

**Fig. 4 | Relapses after platinum exposure in PDX mice are characterized by decreased PHGDH expression and platinum-resistant PDX mice rely on exogenous serine for optimal growth. a** Tumor response to carboplatin treatment of the sensitive OVC1a model and its sensitive relapse OVC1b, OVC1a vehicle-carboplatin, $n = 12–13$ mice, respectively; OVC1b vehicle-carboplatin $n = 10–11$ mice, respectively, repeated measures two-way ANOVA with multiple comparison Tukey's post-test, data are represented as mean ± SEM, $p < 0.0001$.
**b** Representative western blot of PHGDH of OVC1a and OVC1b mice, $n = 3$ and $n = 2$ mice for OVC1a and OVC1b, respectively, and densitometric quantification, western blot was performed three times, unpaired two-tailed $t$-test, data are represented as mean ± SD, $p = 0.0105$. **c** Western blot of $n = 3$ vehicle-treated mice and $n = 3$ relapsed OVC3 mice after two rounds of carboplatin treatment and densitometric quantification of this western blot performed two times ($n = 6$ and $n = 5$ for vehicle and relapse, respectively), unpaired two-tailed $t$-test, data are represented as mean ± SD, $p = 0.0040$. **d** Tumor response to carboplatin treatment and serine/ glycine-free diet in the sensitive OVC1a model, -ser/-gly diet: $n = 10$ vehicle and $n = 10$ carboplatin treated, control diet: $n = 9$ vehicle and $n = 9$ carboplatin treated, repeated measures two-way ANOVA, data are represented as mean ± SEM, $p < 0.0001$. **e** Tumor response to carboplatin treatment and serine/glycine-free diet in the resistant OVC2 model, -ser/-gly diet: $n = 9$ vehicle and $n = 9$ carboplatin treated, control diet: $n = 8$ vehicle and $n = 10$ carboplatin treated, repeated measures mixed-effects two-way ANOVA, data are represented as mean ± SEM, $p < 0.0001$. **f** Total serine levels in OVC1a ($n = 5$ mice) and OVC2 ($n = 4$ mice), unpaired two-tailed $t$-test, data are represented as mean ± SD, $p = 0.0371$. Total serine levels were determined three independent times. **g** Representative western blot of PHGDH of OVC1a ($n = 3$) and OVC2 ($n = 2$) mice and its densitometric quantification, unpaired two-tailed $t$-test, data are represented as mean ± SD, $p = 0.0190$. Western blot was performed three times. Source data are provided as a Source Data file.

affected by the diet (Fig. 4d, e). These data confirm that dependence on exogenous serine for fast proliferation is a metabolic feature of PHGDH-low expressing resistant tumors also in vivo.

## Platinum-resistant cells rewire NAD⁺-regenerating pathways

Although different tumor types have been reported to depend on serine biosynthesis for development and growth[12,23,25], our data suggest that under platinum pressure ovarian cancer cells may gain an advantage by selectively downregulating this process. We therefore sought to investigate how carbon flux through the serine biosynthetic pathway could functionally influence cellular response to platinum in our cells. Since the first reaction catalyzed by the rate limiting enzyme PHGDH consumes NAD⁺ to produce 3-phosphohydroxypyruvate and it is well known that NAD⁺ availability is a major constraint for PHGDH activity (Supplementary Fig. 2d)[32], we hypothesized that resistant cells may reduce carbons flux through serine synthesis at the level of PHGDH to spare NAD⁺. We thus measured NAD⁺/NADH ratio and the levels of the single cofactors in both sensitive and resistant cells and observed a lower NAD⁺/NADH ratio and lower NAD⁺ levels in resistant cells (Fig. 5a, b), with comparable NADH levels (Fig. 5c).

We then integrated our metabolomics data and found that resistant cells show a general metabolic rewiring toward a NAD⁺-regenerating phenotype, in comparison to the naïve sensitive cells (Fig. 5d). While we did not find major differences in oxygen consumption rate (OCR) (Supplementary Fig. 6a), we observed that resistant cells were characterized by lower levels of TCA intermediates (Supplementary Fig. 2c) but sustained pyruvate carboxylase activity under treatment (Supplementary Fig. 6b), probably to replenish them (Supplementary Fig. 6c). We also measured higher lactate and adenosine-5′-triphosphate (ATP) levels (Supplementary Figs. 2c and 6d) and increased lactate secretion (Supplementary Fig. 6e) in resistant cells compared to the sensitive ones. This suggests, in line with the increased glucose uptake (Fig. 2h), an increased glycolytic flux. Although glycolysis continues to use NAD⁺ at the level of glyceraldehyde-3-phosphate-dehydrogenase, our $^{13}C_6$-glucose tracer experiments showed that resistant cells diverted glucose-derived carbons into glycerol-3-phosphate at a higher rate compared to the sensitives ones (Supplementary Fig. 6f, g), thus regenerating NAD⁺ from NADH. In addition, analysis of a $^{13}C_5$-glutamine labeling experiment showed that resistant cells increased reductive carboxylation (Supplementary Fig. 6c, h) as a result of the increased α-ketoglutarate to citrate ratio dictated by the lower NAD⁺ availability (Supplementary Fig. 6i), as reported for cancer cells with impaired mitochondrial activity[33]. It has been recently shown that polyunsaturated fatty acid desaturation is another possible mechanism used by eukaryotic cells to recycle glycolytic NAD⁺ [34]. In support of our hypothesis, we found that resistant cells had increased levels of highly unsaturated fatty acid (Supplementary Fig. 6j), compared to the sensitive ones. Of note, fatty acid elongation is also a source of NADP⁺ and we observed an increase in fatty acid chain length in resistant cells (Supplementary Fig. 6j).

To validate our hypothesis that resistant cells may restrain serine synthesis due to the need of maintaining NAD⁺ availability, we modulated NAD⁺ levels in our resistant cells. Supplementation with α-ketobutyrate (αKB), an electron acceptor that is converted by lactate dehydrogenase to α-hydroxybutyrate thus regenerating NAD⁺ but not contributing carbons to downstream pathways (Fig. 5e)[35], rescued the proliferation of resistant cells under serine/glycine deprivation and carboplatin treatment (Fig. 5f). These data show that increasing NAD⁺/NADH ratio (Fig. 5g) to a certain threshold, by increasing NAD⁺ levels and to a less extent NADH levels (Supplementary Fig. 6k, l), can be enough to overcome the stress induced by carboplatin treatment under serine/glycine deprivation in resistant cells. Importantly, αKB supplementation also led to re-expression of PHGDH (Fig. 5h), suggesting that NAD⁺ availability is a constraint for PHGDH activity in our resistant cells.

Together, the metabolic adaptations observed in resistant cells, schematically illustrated in Fig. 5d, support a scenario in which sensitive cells would funnel most of the glucose into biosynthetic pathways to support their requirements of high proliferation rates. Once exposed to platinum and adapted to grow under its pressure, however, cells switch their central carbon metabolism toward what we speculate could be a NAD⁺-regenerating phenotype, by reducing NAD⁺-using activities such as serine biosynthesis.

Since our findings point to NAD⁺ metabolism as one of the players in adaptations to platinum exposure, we hypothesized that resistant cells could also be susceptible to NAD⁺-depleting strategies. Although both sensitive and resistant cells responded to the Nicotinamide Phosphoribosyltransferase (NAMPT) inhibitor FK866 (APO866, Daporinad) due to the lack of de novo NAD⁺ synthesis[36,37], we found that resistant cells were two times more sensitive to FK866 compared to the sensitive cells, with a GI50 of 0.65 and 1.41 nM, respectively (Fig. 5i). Interestingly, while low-dose treatment with 0.4 nM FK866 as single agent had only a minor effect, in combination with carboplatin it strongly impaired proliferation of resistant cells (Fig. 5j). Moreover, RNA-seq data showed that resistant cells underwent a major reshuffling in NAD⁺-dependent dehydrogenases and other NAD⁺-related enzymes compared to sensitive cells (Supplementary Fig. 7a). In addition, carboplatin treatment altered the expression of these enzymes in sensitive cells, but not in the resistant ones (Supplementary Fig. 7a).

Taken together, our data suggest that resistant cells with downregulated serine biosynthesis also show global metabolic rearrangement toward a NAD⁺-sparing phenotype, and that pharmacological NAD⁺-depleting agents, such as NAMPT inhibitors, may be potentially combined with platinum to overcome resistance.

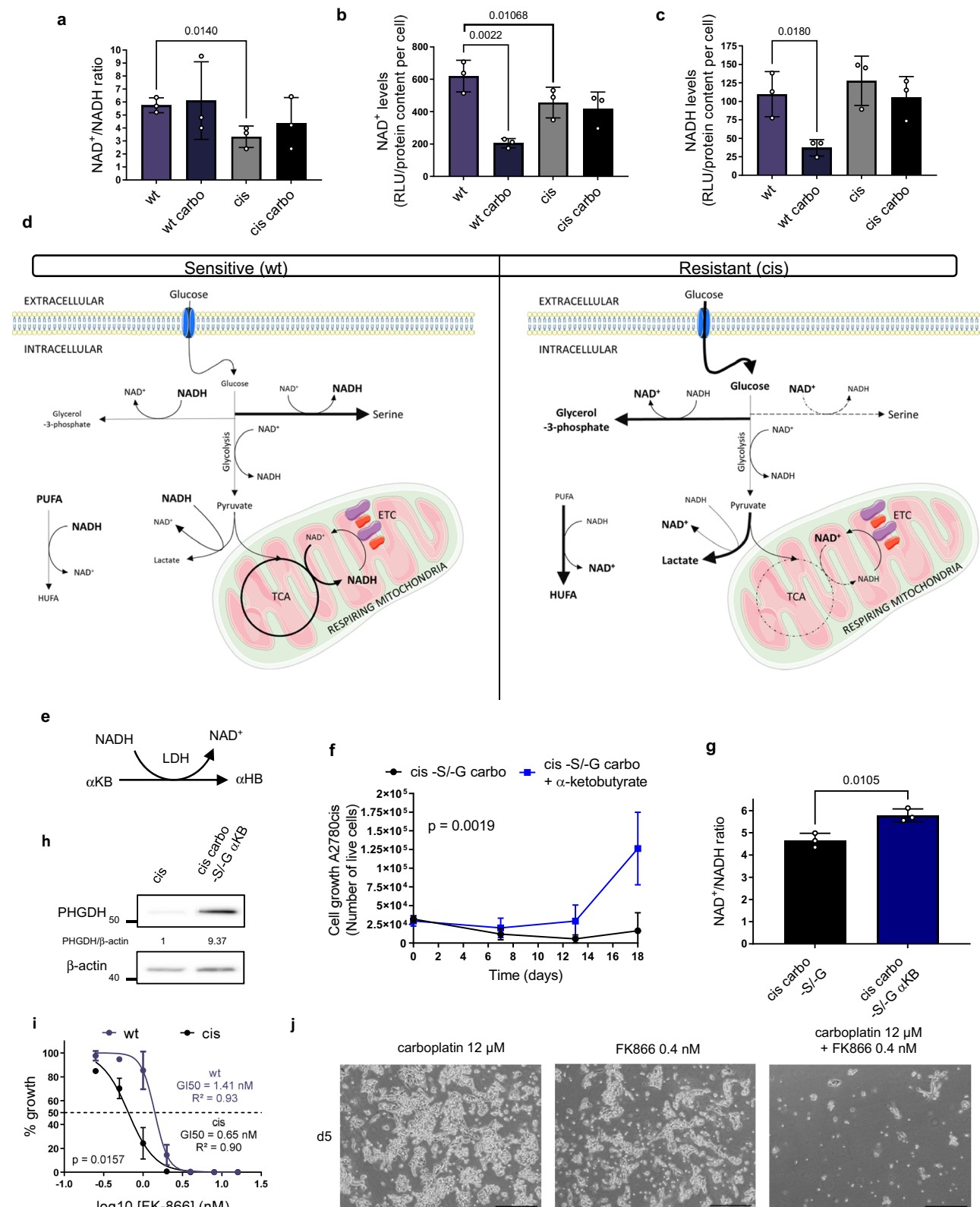

## Resistant cells with decreased serine biosynthetic activity sustain PARP activity under platinum treatment

Because in resistant cells we observed a $NAD^+$-sparing phenotype but not a net increase in $NAD^+$ levels, we hypothesized that resistant cells would reshuffle their $NAD^+$-related metabolism to counteract for excessive $NAD^+$-consumption. PARPs are major $NAD^+$-consuming enzymes involved in DNA repair in eukaryotic cells[38], and A2780

resistant cells are known to have higher ability to tolerate and repair platinum-induced DNA damage, compared to wt cells[39]. A phenotype we confirmed by γH2AX staining and by showing their normal cell cycle profile under treatment (Supplementary Fig 7b, c). Since we observed that resistant cells actively regenerate $NAD^+$ but do not increase their $NAD^+$ levels compared to the sensitive ones, we hypothesized that they may consume more $NAD^+$ through PARP activation

**Fig. 5 | Resistant cells rewire NAD⁺-regenerating pathways.** NAD⁺/NADH ratio (**a**), NAD⁺ levels **b** and NADH levels (**c**) determined by NAD⁺/NADH-Glo bioluminescent Promega assay, $n = 3$ biological replicates (with three technical replicates each time), unpaired two-tailed $t$-tests, data are represented as mean ± SD. **d** Metabolic map and observed alterations of wt vs. cis cells, more elaborate results in Supplementary Fig. 6. **e** The use of α-ketobutyrate (αKB) by lactate dehydrogenase (LDH) to generate α-hydroxybutyrate (αHB) and NAD⁺. **f** 2 mM αKB supplementation of serine/glycine starved cis cells with 6 μM carboplatin treatment, $n = 3$ biological replicates (with three technical replicates each time), repeated measures two-way ANOVA, data are represented as mean ± SD, $p = 0.0019$. **g** NAD⁺/NADH ratio of αKB supplemented serine/glycine starved cis cells with carbo treatment, $n = 3$ biological replicates (with three technical replicates each time), unpaired two-tailed $t$-test, data are represented as mean ± SD, $p = 0.0105$. **h** Representative western blot of PHGDH in αKB supplemented carboplatin treated ser/gly starved cis cells, $n = 3$

biological replicates. **i** GI50 of wt and cis cells to the NAMPT inhibitor Daporinad determined with trypan blue counting, $n = 4$ biological replicates (with three technical replicates each), unpaired two-tailed $t$-test between GI50 values, data are represented as mean ± SEM, $p = 0.0157$. **j** Representative images of cis cells under 12 μM carboplatin and 0.4 nM Daporinad treatment and its combination, scale bar is 500 μm, $n = 3$ biological replicates (with three technical replicates each). Some schematic art pieces in **d** were used and modified from Servier Medical Art. Servier Medical Art by Servier is licensed under a Creative Commons Attribution 3.0 Unported License (https://creativecommons.org/licenses/by/3.0/). Source data are provided as a Source Data file. Carbo carboplatin treated, αKB α-ketobutyrate, αHB α-hydroxybutyrate, LDH lactate dehydrogenase, ETC electron transport chain, TCA trycarboxylic acid cycle, PUFA polyunsaturated fatty acids, HUFA highly unsaturated fatty acids.

and thus we measured levels and activity of PARP1, the most abundant member of the PARP family involved in DNA damage. We observed that resistant cells expressed higher PARP1 levels compared to the sensitive ones (Fig. 6a), and showed evidence of sustained PARP activity, measured as histone PARylation capacity, under carboplatin treatment (Fig. 6b). Interestingly, we observed a significantly decrease in PARylation in resistant cells overexpressing PHGDH (Fig. 6c), and that our serine synthesis-reactivated cis SSP cells decreased PARP1 expression again, while increasing PHGDH levels and carboplatin sensitivity (Figs. 3a, b, e and 6d). This suggests that in our resistant cells PHGDH and PARP could compete for NAD⁺ availability. In line, using ovarian cancer cell line data from the Cancer Cell Line Encyclopedia (CCLE) and Genomics of Drug Sensitivity Consortium (GDSC)[40], we found that PHGDH-low expressing ovarian cancer cell lines were more resistant to the PARP inhibitor olaparib (Fig. 6e). We then investigated the effect of pharmacological PARP inhibition in resistant cells exposed to acute platinum treatment. While resistant cells had a higher GI50 value for the PARP inhibitor olaparib compared to sensitive cells (Fig. 6f), likely due to the observed higher PARP activity (Fig. 6a, b), we observed an additive effect of the PARP inhibitors in combination with carboplatin in decreasing the proliferation of resistant cells (Fig. 6g, h). We then tested the same combination treatment strategy in an additional in vitro model, the OVCAR3 cell line, originally derived from ovarian cancer ascites isolated from a HGSC patient refractory to cisplatin, cyclophosphamide and Adriamycin treatment, also commonly used to investigate platinum resistance[41]. Although in vitro we observed a higher sensitivity to carboplatin compared to the A2780 resistant cells (Supplementary Fig. 7d), we confirmed also in OVCAR3 that combination of sub-lethal doses of carboplatin and olaparib impaired growth and induced cell death (Fig. 6i). Of note, olaparib as monotherapy had limited effect on both cell lines, confirming our hypothesis that sustaining PARP activity may play a role in survival of resistant cells specifically under carboplatin pressure.

We also tested the possibility of combining olaparib and FK866 on resistant A2780cis cells and OVCAR3 cells, and we found that the combination can indeed slow down cell growth in both cell lines, however the effect was milder compared to the carboplatin-olaparib combination (Supplementary Fig. 7e, f).

Taken together, our data show that the observed NAD⁺-regenerating phenotype in resistant cells is associated with sustained PARP activity and sustains cell survival upon acute platinum treatment.

## PARP inhibition re-sensitizes organoids derived from resistant PDX models to carboplatin

Since low serine biosynthetic activity and increased dependence on exogenous serine seemed to be peculiar vulnerabilities of platinum-resistant tumors also in vivo, we tested the susceptibility of two additional HGSC cancer PDX models named OVC4 and OVC5 to PARP inhibition combined with carboplatin, using PDX-derived organoid cultures (Fig. 7a–d). OVC4 has been established from a patient who

developed resistance several years after diagnosis while the OVC5 model has been established from a mucinous ovarian cancer patient with an acquired resistant tumor (Supplementary Table 3). First we analyzed both organoid models by H&E staining and PAX8 and mutant p53 IHC, and we matched with corresponding PDX models (Supplementary Fig. 8a, b). OVC5 organoid cultures were more resistant to carboplatin than OVC4 cultures as shown by their GI50 values (16.10 μM vs. 7.59 μM and 4.12 μM, respectively, Supplementary Fig. 8c). The treatment experiment on both organoid cultures confirmed our findings in the cell lines, with the combination significantly affecting cancer growth and almost no effect for any of the single treatments (Fig. 7a–d). These data validate our in vitro findings in clinically relevant ex vivo samples, and suggest that at least a fraction of resistant tumors might be re-sensitized to platinum treatment by inhibiting their PARP dependent activity.

Since the use of PARP inhibitors is currently indicated for platinum sensitive relapses, and as maintenance after platinum for platinum sensitive patients with homologous recombination (HR) deficient tumors, we checked the mutational status of several HR-related genes in our different models. To do this, we performed whole exome sequencing (WES) on the A2780wt/cis and OVCAR3 cell lines. For all three cell lines we found a very low total mutational burden and only limited mutations in HR genes (Supplementary Fig. 8d). The same analysis for all PDX models and their derived organoids showed higher variability in total mutational burden and HR-related mutations (Supplementary Fig. 8e, f), suggesting that our findings about the potential to combine carboplatin with PARPi is independent of HR status of the different models. This could be explained by the fact that different tumors might rely on different PARP activities, spanning from DNA damage response to transcriptional regulation.

## Discussion

Development of resistance to platinum-based chemotherapy is a major hurdle in the treatment of ovarian cancer and a prominent cause of its poor prognosis. While elevated serine synthesis is linked to high proliferation rates, aggressiveness and ultimately poor prognosis in different tumors[12,23,25,42], a recent report showed that its inhibition leads to increased cisplatin resistance in gastric cancer cells in vitro[43]. However, the functional and clinical relevance of serine synthesis downregulation in platinum resistance remains elusive, as well as its potential clinical exploitability. Here we show that serine biosynthesis downregulation is one of the major adaptations that a fraction of resistant cells can use to preserve intracellular NAD⁺ levels, in line with the recent report that NAD⁺ and NADH abundances are major constraints for serine synthesis in different cancer cell types[32,44]. The compelling hypothesis that PHGDH and PARP1 could compete for NAD⁺ availability under platinum-induced stress is not surprising in light of the recent discovery that under metabolic stress the concomitant presence of both enzymes in the nucleus can create local NAD⁺ gradients, thus influencing PARP-mediated transcriptional activities[45].

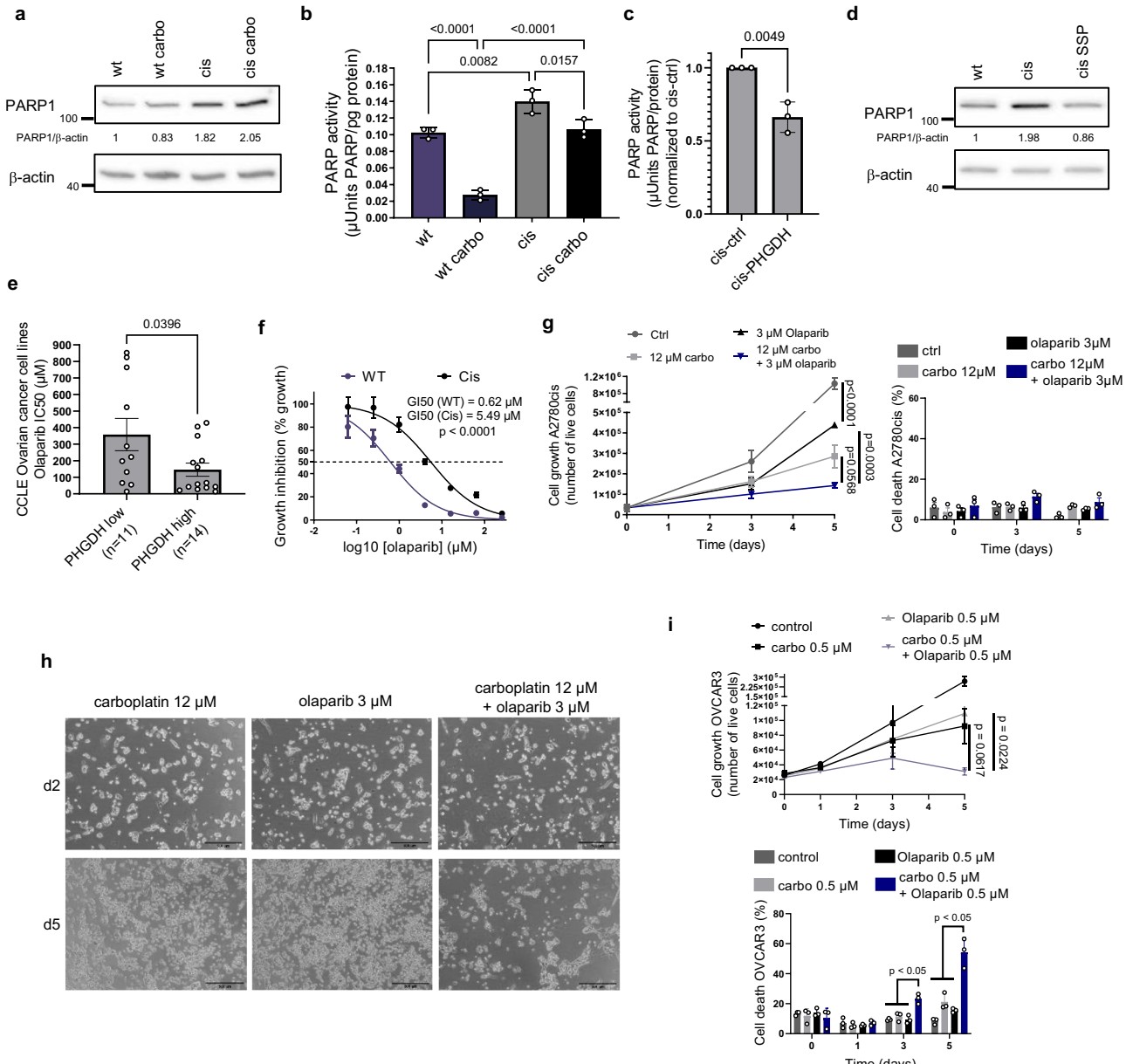

**Fig. 6 | Resistant cells with decreased serine synthesis sustain PARP activity under carboplatin treatment. a** Western blot of PARP1 in wt and cis cells, $n = 4$ biological replicates. PARP activity in carboplatin treated wt and cis cells (**b**) and in cis-PHGDH overexpressing cells (**c**), $n = 3$ biological replicates (with three technical replicates each), ordinary one-way ANOVA with Tukey's multiple comparison post-test, data are represented as mean ± SD with $p < 0.0001$ (**b**) and $n = 3$ biological replicates (with three technical replicates each), unpaired two-tailed $t$-test, data are represented as mean ± SD, $p = 0.0049$ **c. d** Western blot of PARP1 in cis SSP cells, $n = 2$ biological replicates. **e** Olaparib sensitivity of PHGDH[low] and PHGDH[high] expressing ovarian cancer cell lines, data obtained from the CCLE (gene expression) and GDSC (drug sensitivity), $n = 25$ divided in PHGDH[low] (PHGDH mRNA $Z < 0$, $n = 11$) and PHGDH[high] (PHGDH mRNA $Z > 0$, $n = 14$) subgroups, unpaired two-tailed $t$-test, error bars represent SEM, $p = 0.0396$. **f** GI50 value of wt and cis cells for the PARP inhibitor olaparib, $n = 3$ biological replicates (with three technical replicates each),

unpaired two-tailed $t$-test between GI50 values, data are represented as mean ± SD, $p < 0.0001$. Growth and cell death of cis cells with 12 µM carboplatin and 3 µM olaparib and its combination, $n = 3$ biological replicates (with 2–3 technical replicates each) (left panel is a representative figure with $n = 2$ technical replicates), two-way ANOVA with Tukey's multiple comparison post-test at day 5, data are represented as mean ± SEM with $p < 0.0001$ (**g**) and its representative images, scale bar is 500 µm, $n = 3$ **h. i** Growth (left) and cell death (right) of OVCAR3 cells under 0.5 µM carboplatin and 0.5 µM olaparib and its combination, $n = 3$ biological replicates (with three technical replicates each), unpaired two-tailed $t$-test at day 5 with $p = 0.0617$ and $p = 0.0224$ for carbo + olaparib vs carbo or olaparib, respectively (left) and two-way ANOVA with Tukey's multiple comparison post-test with $p = 0.0424$ (right), data are represented as mean ± SEM. Source data are provided as a Source Data file. Carbo carboplatin treated, CCLE Cancer Cell Line Encyclopedia, GDSC Genomics of Drug Sensitivity in Cancer.

However, serine biosynthetic activity and serine dependency have never been associated with platinum response in ovarian cancer before. Our data in PDX models and the analysis of matched clinical samples collected at diagnosis and at relapse also show that this phenotype may be acquired gradually in a subset of patients, due to subsequent exposure to platinum-based therapy cycles. This is particularly relevant, since most of the patients experience several response-relapse-response events during the course of their disease. Although further studies comparing matched biopsies prospectively and longitudinally collected from larger series of patients are needed to validate these findings, our results suggest that serine biosynthetic activity, and specifically PHGDH and intratumor serine levels, could

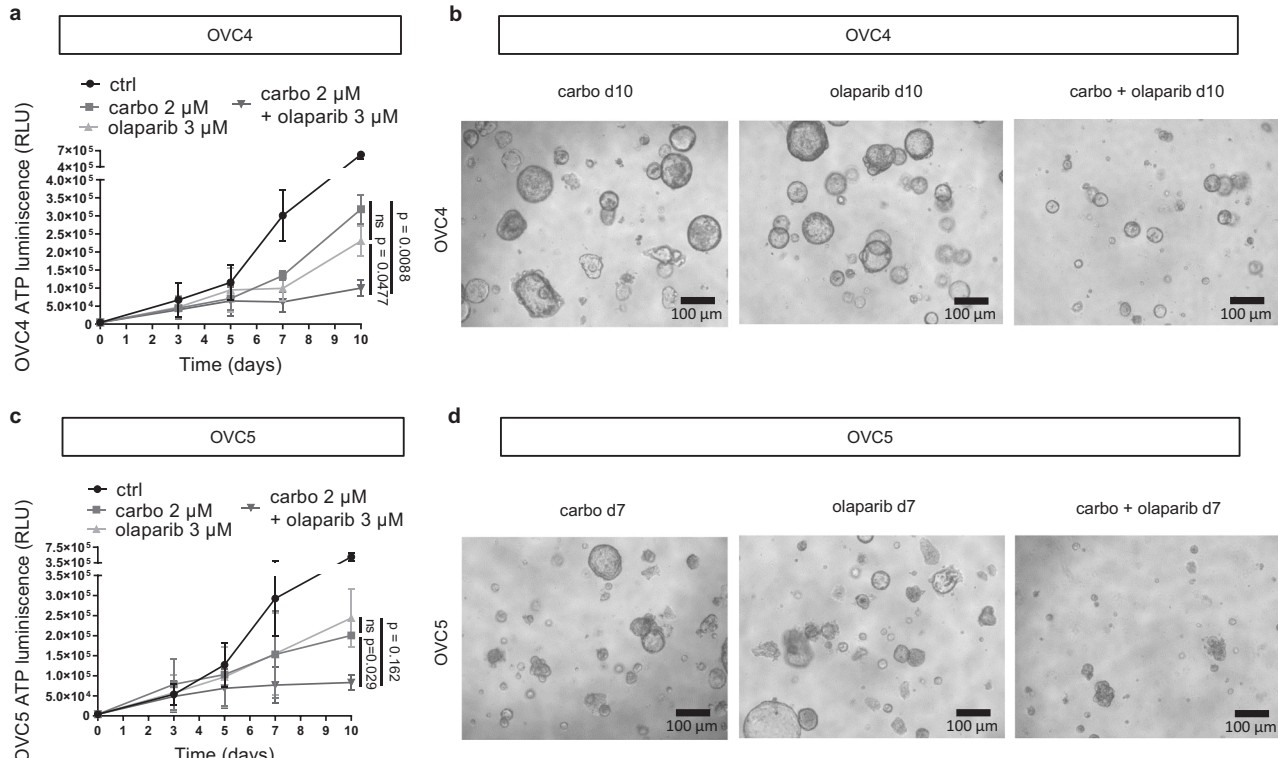

**Fig. 7 | PARPi re-sensitizes resistant PDX-derived organoids to carboplatin.**
Growth of **a** OVC4 organoids under carboplatin and olaparib single agent treatment and its combination, determined by adenosine triphosphate (ATP) measurement, $n = 3$ biological replicates (with six technical replicates each), data are represented as mean ± SEM, unpaired two-tailed $t$-test performed at day 10, and **b** its representative pictures, scale bar is 100 μm. Growth of **c** OVC5 organoids under carboplatin and olaparib single agent treatment and its combination, determined by ATP measurement, $n = 2$ biological replicates for treatment groups and $n = 3$ for ctrl growth (with six technical replicates each), data are represented as mean ± SEM, unpaired two-tailed $t$-test performed at day 10, and **d** its representative pictures, scale bar is 100 μm. Source data are provided as a Source Data file. Carbo carboplatin treated organoids.

potentially predict development of platinum resistance in a subgroup of patients.

Our findings have twofold implications, shedding light on mechanisms underlying the multiple adaptions described so far for different pathways involved in glucose, glutamine and amino acids metabolism in platinum-resistant ovarian cancers, but also revealing unexpected vulnerabilities that can potentially be exploited to re-sensitize resistant cells[19,46]. Specifically, our data suggest that both PARP and NAD+ biosynthesis inhibition, in combination with platinum-based chemotherapy, could be considered for patients with resistant disease that decrease serine biosynthesis. The original approval of three different PARP inhibitors (olaparib, niraparib and rucaparib) as maintenance therapy for recurrent platinum sensitive patients with gBRCA mutations marked an important advance in the management of ovarian cancer patients[47–49]. Clinical evidence shows that also patients without BRCA mutation or homologous recombination deficiency (HRD) could benefit from PARP inhibitors and this might not be limited to the maintenance setting[48,50]. Intriguingly, mutational analysis of our preclinical models suggests that response to carboplatin-olaparib combination in resistant patients seem to be HRD-independent. However, a systematic definition of the HR status in research subjects is needed to be able to draw firm conclusions, highlighting the importance of future prospective studies in HR status-defined patients sampled longitudinally after platinum treatments. In this view and considering the low expected response rate to other chemotherapies in resistant patients (10–15%), the possibility to at least temporarily control the progression of tumors by combining platinum and PARP inhibitors is appealing. In addition, since in the imminent future PARP inhibitors will be used also in earlier clinical settings and patients will be exposed to such drugs for longer periods, the adoption of NAD+

synthesis inhibitors could potentially overcome development of acquired resistance to PARPi. Further proof-of-concept studies are needed to clinically validate our preliminary findings. While open questions remain, not the least if our findings and the metabolic vulnerabilities we uncovered are specific for ovarian cancer, our data provide a rationale for future studies to assess the relation between serine biosynthesis and platinum resistance in ovarian cancer and additional tumor types treated with platinum-based chemotherapy.

## Methods
### Cell culture
A2780wt and its cisplatin resistant analog A2780cis cells were obtained from the ECACC (Sigma-Aldrich ECACC 93112519 and Sigma-Aldrich ECACC 93112517, respectively) and were maintained in RPMI medium containing 10% FBS (Gibco), 2 mM glutamine (L-Glutamine, Gibco), 100 IU penicillin and 100 μg/ml streptomycin (PenStrep, Gibco). OVCAR3 cell line was purchased from the ATCC (NIH:OVCAR-3 (ATCC® HTB-161™)) and were also grown in RPMI medium containing 10% FBS, glutamine and PenStrep. They were authenticated using short tandem repeat profiling. HEK293T cells were obtained from ATCC (CRL-3216™) and were grown in DMEM containing 10% FBS (Gibco), 2 mM glutamine (L-Glutamine, Gibco), 100 IU penicillin and 100 μg/ml streptomycin (PenStrep, Gibco). Cell lines were routinely tested for Mycoplasma contamination and they were generally used for experiments between passages 10 and 25.

### Chemotherapeutic agents and chemicals
Carboplatin was provided by the university hospital Leuven (UZL), who purchased it from Hospira. NAC, αKB, FK866 (Daporinad), L-$^{13}$C$_5$-glutamine and L-$^{13}$C$_3$-serine were obtained from Sigma-Aldrich, whereas

D-glucose $^{13}C_6$ was purchased from Cambridge Isotope Laboratories. Olaparib (AZD2281) was purchased from Selleckchem. All ingredients to make fresh RPMI were purchased from Sigma-Aldrich. Hygromycin B was purchased from Invitrogen, G418 sulfate (Geneticin) and puromycin were purchased from Gibco.

## Antibodies

PHGDH (HPA021241, 1:2000 for western blot, 1:4000 or 1:8000 for IHC for the Norwegian and Dutch cohort, respectively) and PSPH (HPA020376, 1:1000 for western blot) antibodies were purchased from Sigma-Aldrich. The PSAT1 (CPTC-PSAT1-2, 1:500 for western blot) and PARP1 (AFFN-PARP1-17B10, 1:150 for western blot) antibodies, developed by the National Cancer Institute and EMBL MACF, respectively, were obtained from the Developmental Studies Hybridoma Bank, created by the NICHD of the NIH and maintained at The University of Iowa, Department of Biology, Iowa City, IA 52242. The phospho-Histone H2A.X (Ser139) (20E3, 1: 600 for IF), ATF4 (D4B8, 1:1000 for WESTERN BLOT) and β-actin (13E5, 1:5000, for western blot) antibodies were purchased from Cell Signaling Technology. Anti-p53 Antibody [DO-1]-Chip graded was obtained from Abcam (ab1101, 1:200 for IHC) while PAX8 Polyclonal antibody was obtained from Proteintech (10336-1-AP, 1:1200 for IHC). Secondary peroxidase conjugated goat anti-rabbit (111-035-003, 1:5000 for western blot (1:10 000 for β-actin western blot)) and goat anti-mouse antibodies (115-035-044, 1:10,000 for western blot) were purchased from Jackson ImmunoResearch. Secondary donkey anti-rabbit Alexa Fluor 594 (1:800) was purchased from Molecular Probes, Life Technologies.

## Cell growth and death assay

Cell proliferation and death were analyzed by counting total cell number with a hemocytometer after staining with trypan blue. Medium was changed on day 0 and 2 and every other 3 days after. Cell numbers were determined at time points indicated in figures. All experiments were performed at day 5, except if stated otherwise. For experiments under serine/glycine deprivation conditions, cells were fed homemade medium (homemade RPMI with the same formulation as commercial RPMI from Gibco, supplemented with 10% dialyzed FBS (Gibco) and standard concentrations of penicillin-streptomycin and glutamine) with or without serine and/or glycine (Sigma-Aldrich).

## Genetic overexpression of catalytically-active PHGDH and catalytic-dead PHGDH (CD-PHGDH)

Overexpression of PHGDH was performed as described before[26]. The overexpression of PHGDH in A2780cis cells was performed using the pLHCX plasmid expressing the PHGDH cDNA obtained from the lab of S.M.F. The empty plasmid pLHCX was used as a control. Retroviral particles were produced using the pCL Ampho retroviral packaging plasmid in HEK293T cells. Infection of A2780cis cells was performed overnight (ON) with freshly prepared virus and the medium was replaced the next day. Cells were selected with hygromycin (400 μg/ml for initial selection, 200 μg/ml as maintenance concentration). Lentiviral overexpression of CD-PHGDH was performed using pCW plasmid expressing codon optimized catalytically inactive PHGDH (D175N, R236K, H283A). pCW-codon optimized catalytically inactive PHGDH was a gift from Michael Pacold (Addgene plasmid # 154903; RRID:Addgene_154903)[51]. The empty pCW57-MCS1-P2A-MCS2 (Neo) plasmid was used as control. pCW57-MCS1-P2A-MCS2 (Neo) was a gift from Adam Karpf (Addgene plasmid # 89180; RRID:Addgene_89180). Lentiviral particles were produced using the pCMV delta R8.91 and pCMV-VSV-G plasmids (packaging and envelope plasmids, respectively) in HEK293T cells. Infection of A2780cis cells was performed ON with freshly prepared virus and the medium was replaced the next day. Cells were selected with Geneticin (800 μg/ml for initial selection, 500 μg/ml as maintenance concentration). Overexpression of (CD-) PHGDH was validated by western blot analysis.

## Knock-down of PHGDH

PHGDH was knocked-down in platinum sensitive cells using two commercially available shRNAs targeting PHGDH (TRCN0000255351 and TRCN0000233029), using MISSION shRNA lentiviral transduction particles (Sigma). Mission pLKO.1-puro Non-target shRNA control transduction particles (SHC016V-1EA, Sigma) were used as control. In total, 150,000 cells were seeded per well in a 12-well plate. The next day, the lentivirus was put on the cells with a MOI of 0.5 in the presence of 5 μg/ml polybrene. After 24 h, medium was replaced with prewarmed complete growth medium. On day 3, infected cells were selected and further grown in the presence of 2 μg/ml puromycin (Gibco).

## Microscopy

For light microscopy, pictures were taken with a Leica DFC295 (for cells) or Leica DM5500 (for tissue) microscope and Leica Application Suite Version 3.7.0 or LAS AF imaging software, respectively (Leica Microsystems CMS GmBH). Immunofluorescence images were acquired with a fluorescence Olympus BX41 microscope with XC30 camera and X-cite series 120 Q lamps. Using cellSens Dimension software. Analysis of microscopic pictures was done using ImageJ.

## Immunofluorescence

Cells were fixed with 4% formaldehyde for 15 min. Next, specimens were blocked with 10% normal donkey serum (Sigma)/0.3% Triton X-100 in 1X PBS (Gibco) for 1 h and incubated ON at 4 °C with primary antibody. Afterwards cells were incubated for 45 min in the dark with donkey anti-rabbit Alexa Fluor 594. Finally, slides were mounted with Prolong Gold antifade reagent with DAPI (Molecular Probes P36935, Thermo Fisher) and visualized with fluorescence microscopy.

## Cell cycle analysis

Cell cycle phases were analyzed using dual Click-iT EdU Alexa Fluor 488 and FxCycle Violet staining (Molecular Probes, Life Technologies) according to the manufacturer's protocol and analyzed using flow cytometry (BD FACSCanto II, BD Biosciences).

## Oxygen consumption

OCR was determined using a Seahorse XF24 Extracellular flux analyzer (Seahorse Bioscience). In total, 150,000 cells per well were seeded in a XF24 Cell Culture Microplate (Seahorse Bioscience) 24 h before the measurements. OCR was measured under basal conditions and after injection with oligomycin, FCCP and antimycin A, as described before[52].

## Western blot

Cells were lysed in Pierce RIPA Buffer (Thermo Fisher) supplemented with phosphatase and protease inhibitors (Roche-04906845001, Roche-04698116001). Protein concentrations were determined using Pierce BCA Protein Assay Kit (Thermo Fisher) and detected with a Multiskan spectrophotometer (Thermo Fisher). In total, 20 μg of protein was loaded in each lane, except in Fig. 4h where 6 and 36 μg protein were loaded for OVC1a and OVC2, respectively, due to the inherent different nature of the tumors. Before running, protein sample was reduced in reducing buffer (50 mM DTT, 2x Laemmli sample buffer (BioRad)) and heated at 95 °C for 5 min and then separated on a 10-well Mini-PROTEAN TGX Precast Any-KD gel (BioRad) for 10 min at 100 V and subsequent ~40–50 min at 160 V. Gels were blotted onto a Mini Trans-Blot nitrocellulose membrane (BioRad) using 300 mA for 1 h, according to manufacturer's protocol. The membrane was stained with 0.1% Ponceau-Xyline in 5% acetic acid (Sigma) for 1 min to visualize blotted protein bands on the membrane. Membrane was cut into pieces when needed, using the Spectra™ Multicolor Broad Range Protein Ladder (Thermo) as reference. After washing, the membrane was blocked for 1 h at room temperature in 10% fat-free milk in TBS-T

buffer, incubated with primary antibodies with appropriate dilution (see "antibodies") at 4 °C ON and subsequently with secondary antibody for 1 h at room temperature. Bands were visualized using SuperSignal West Dura Extended Duration Substrate (Thermo Fisher) and images were captured using an Azure c600 detection system (Azure Biosystems). Densitometric quantification was performed using ImageJ.

## ATP, and NAD$^+$ and NADH level determination

ATP was measured using ATP-lite assay, NAD$^+$ and NADH levels and ration were determined using NAD$^+$/NADH-glo assay. All assays were purchased from Promega and experiments were executed according to manufacturer's protocol. Bioluminescence was measured with Victor X4 plate reader (PerkinElmer).

## PARP activity assay

PARP activity was determined as PARylation levels in cell extracts on a histone coated 96-well plate. It was measured using the HT Colorimetric PARP/Apoptosis Assay (Trevigen) and experiments were executed according to manufacturer's protocol. Absorbance was detected with a Multiskan spectrophotometer (Thermo Fisher).

## Oxidative stress detection

Mitochondrial superoxide production was detected by measuring MitoSOX (Thermo Fisher, M36008) fluorescence while CellROX Deep Red Reagent (Thermo Fisher, C10422r) was used to measure total cellular ROS with flow cytometry (BD FACSCanto II, BD Biosciences), according to manufacturer's protocol.

## H&E staining

Four μm thick FFPE slides were deparaffinized with toluol (2 × 5 min) and ethanol (2 × 5 min). Slides were incubated in haematolylin solution (Gill3, Prosan) for 4 min. Afterwards slides were dipped three times in 1% HCl in technical ethanol and dipped three times in saturated Li$_2$CO$_3$ solution. Slides were stained for 3 min using eosine and dehydratation was done using propanol (3 times 3 dips) and xylol (3 times 3 dips). Between different steps, slides were washed with tap water and deionized water, respectively. Finally, slides were covered using Depex and covered glass.

## P53 and PAX8 IHC

PAX8 and p53 IHC on organoid FFPE slides were done by routine IHC procedure. In brief, slides were dewaxed in xylene (PAX8)/toluol (p53) and rehydrated in graded ethanol solution. Endogenous peroxidase activity was blocked by 0.3% H$_2$O$_2$ in methanol solution for 30 min. Heat-induced epitope retrieval was done using sodium citrate buffer (10 mM, pH = 6) at 95 °C for 30 min (PAX8) or 1 h (p53). Slides were cooled down to room temperature for 20 min, after which they were blocked for 1 h at room temperature in blocking buffer (2% BSA, 1% dry milk, 0,10% Tween20) + 10% goat serum (for PAX8). Next, sections were incubated overnight at RT (PAX8) or 4 °C (p53) with primary antibody in a humid chamber. Primary antibodies were diluted as described in "Antibodies" in blocking buffer + 1% goat serum (for PAX8). The following day slides were incubated and detected using Envision+/HRP mouse reagent (K400111-2, Agilent) for 30 min for p53. PAX8 slides were incubated with HRP-conjugated secondary antibody for 30 min at room temperature. Visualization was done by applying DAB as chromogenic substrate. Finally, slides were dehydrated in ethanol followed by xylene and sections were mounted with a cover slide using DPX.

## Isotope tracer experiments and GC-LC/MS analysis

Cells were seeded in 6-well plates at a density of 50,000 cells per well in complete medium for proliferation and metabolite assays (day −1). The following day, cells were grown in medium with or without 6 μM carboplatin. At day 2, medium was aspirated, cells washed with PBS, and fresh glucose- or glutamine free RPMI (Gibco) supplemented with $^{13}C_6$-glucose (CLM-1396-5, Cambridge Isotope Laboratories, Inc.) or $^{13}C_5$-L-glutamine (605166, Sigma-Aldrich), respectively, 10% dialyzed FBS, 2 mM glutamine, 100 IU penicillin, 100 μg/ml streptomycin, with or without 6 μM carboplatin, was added. At day 5, cells were washed with 0.9% NaCl, and quenched by placing the plate in liquid nitrogen and stored at −80 °C. Alternatively, quenching and metabolite extraction was also performed after 24 h.

Metabolite extraction and detection by gas chromatography/mass spectrometry or liquid chromatography/mass spectrometry was performed as described before[26,53]. Metabolites were extracted with 800 μl of MeOH/H$_2$O (5:3) (v:v) containing 0.6 μg/ml of glutaric acid on a dry ice/ice mixture, scraped, transferred to 2-ml tube, vortexed (10 min, 4 °C) and centrifuged (10 min, 20,000 × $g$, 4 °C). The supernatants were dried ON under vacuum at 4 °C.

**Gas chromatography/mass spectrometry (GC/MS).** Samples were derivatized using methoxyamine hydrochloride and *N*-methyl-*N-tert*-butyldimethylsilyltrifluoroacetamide (TBDMS). Samples were dissolved in 20 μl of 20 mg/ml methoxyamine hydrochloride in pyridine (MOX) (Pierce) at 37 °C for 90 min. Samples were centrifuged for 5 min, and 7.5 μl of supernatant was transferred to a glass vial. Samples were then derivatized by adding 15 μl TBDMS + 1% tert-butyldimethylchlorosilane (TBDMCS; Pierce) for 1 h at 60 °C. Analysis was performed on an Agilent 7890A GC system coupled to an Agilent 5975C Inert MS system with an electron impact ionization set at 70 eV operating in SIM mode. Metabolites were separated with a DB35MS column (30 m, 0.25 mm, 0.25 μm) using helium carrier gas at a flow rate of 1 ml/min. A volume of 1 μl of sample were injected with a split ratio 1 to 3 with an injector set at 270 °C. For metabolite separation, the GC oven temperature was set at 100 °C for 1 min ramped to 105 °C at 2.5 °C/min, then to 240 °C at 3.5 °C/min and finally to 320 °C at 22 °C/min. Data were collected by Masshunter version 10.1 build 10.1.48 (Agilent). Fractional enrichment of $^{13}$C in metabolites was corrected for the natural abundance using an in-house script written in Matlab using the method developed by Fernandez et al.[54] Metabolites abundances were also normalized to the internal standard glutaric acid.

**Liquid chromatography/mass spectrometry (LC/MS).** For the detection of metabolites by LC-MS, a Dionex UltiMate 3000 LC System (Thermo Scientific) with a thermal autosampler set at 4 °C, coupled to a Q Exactive Orbitrap mass spectrometer (Thermo Scientific) was used. Samples were resuspended in 50 μl of water and a volume of 10 μl of sample was injected on a C18 column (Acquity UPLC HSS T3 1.8 μm 2.1 × 100 mm). The separation of metabolites was achieved at 40 °C with a flow rate of 0.25 ml/min. A gradient was applied for 40 min (solvent A: 10 mM Tributyl-Amine, 15 mM acetic acid−solvent B: Methanol) to separate the targeted metabolites (0 min: 0% B, 2 min: 0% B, 7 min: 37% B, 14 min: 41% B, 26 min: 100% B, 30 min: 100% B, 31 min: 0% B; 40 min: 0% B). The MS operated in negative full scan mode (*m/z* range: 70−1050 and 300−700 from 5 to 25 min) using a spray voltage of 4.9 kV, capillary temperature of 320 °C, sheath gas at 50.0, auxiliary gas at 10.0. Data were collected using the Xcalibur software v4.0 (Thermo Scientific) and analyzed with Matlab for the correction of protein content and natural abundance, but also to determine the isotopomer distribution using the method developed by Fernandez et al.[54].

## Lipid profiling

Cells were collected at day 5 and dissolved in 800 μl DPBS. In total, 100 μl of the suspension was used for protein determination using the BCA assay (Thermo Fisher). The remaining suspension was used for lipid extraction as described before[55]. Samples were homogenized in 0.9 ml 1 N HCl/CH3OH ULC/MS (1:8, v/v) (Biosolve), 0.8 ml CHCl3

(Sigma-Aldrich) and 200 μg/ml of the antioxidant 2,6-di-tert-butyl-4-methylphenol (BHT, Sigma). The suspension was shaken for 10 min and spun (SS-34, 12,000 rpm for 5 min at 4 °C with the brake ON). The lower phase was collected to glass tubes and evaporated under vacuum (Speed Trap, Thermo Fisher) for 30 min at room temperature. Samples were sparged with Argon (Sigma) and stored at −20 °C. At the day of analysis, the samples were reconstituted in CH3OH/CHCl3/NH4OH (72:8:1, v/v/v) and lipid standards were added (Avanti Polar Lipids). Phospholipids were analyzed by electrospray ionization tandem mass spectrometry (ESI-MS/MS) with a hybrid quadrupole linear ion trap mass spectrometer (4000 QTRAP system, AB SCIEX) equipped with a TriVersa NanoMate robotic nanosource (Advion Biosciences). The collision energy was varied as follows: prec 184, 50 eV; nl 141, 35 eV; nl 87, −40 eV; prec 241, −55 eV. The system was operated in the multiple reaction monitoring mode for quantification of individual species. Typically, a 3-min period of signal averaging was used for each spectrum.

## RNA-seq analysis

In total, $5 \times 10^6$ cells per condition per replicate were lysed using RLT buffer (Qiagen) and stored at −80 °C. RNA-seq was performed using HiSeq2500 platform (Illumina) (SR-65bp) at the Genomics Core facility of the NKI-AvL (Amsterdam, the Netherlands). Raw RNA-seq FASTQ files were aligned with STAR to the human genome GRCh37.p13. The file formats were then converted using SAMtools. Reads were counted at exons of protein-coding genes using HTSeq with gencode.v19 annotation. Normalization of library size for the read counts and differential expression analysis was performed using DEseq and DEseq2 with standard parameters. RNA-seq data have been deposited into the NCBI Gene Expression Omnibus database (http://www.ncbi.nlm.nih.gov/geo/) under accession number GSE176218.

## Whole exome sequencing (WES)

Genomic DNA of cells, organoids and tumors was extracted using the DNeasy Blood & Tissue kit (Qiagen) according to the manufacturer's protocol. WES was performed at the UZ Leuven Genomics Core. For WES, libraries were made on Sciclone (PerkinElmer, USA) using KAPA HyperPrep kit (Roche, WI, USA) and IDT Dual Indexed Adapters (IDT, USA). Exons were enriched through hybridization with a custom version of the in-solution Twist Biosciences Human Comprehensive Exome kit. The enriched libraries were then amplified and sequenced on NovaSeq 6000 as Paired end 150 bp (Illumina, CA, USA). WES were processed using an in-house pipeline. Base-calling was performed with bcl2fastq (version 2.19.0). Reads were aligned to the human reference genome GRCh38 using BWA mem (version 0.7.17), duplicate reads were marked using Picard MarkDuplicates (version 2.22.1) and base quality scores were recalibrated using GATK BaseRecalibrator and ApplyBQSR (version 4.1.7). Finally, single nucleotide variants and small indes were called using GATK HaplotypeCaller (version 4.1.7) and annotated using Annovar (version 2019Oct24). Variant filtering was performed in R based on quality score, coverage and pathogenicity score. Variants with a Phred corrected quality score of <50, a coverage of <10x and >243 and Phred corrected CADD pathogenicity score <15 were removed. Variants that met these requirements were incorporated into the final mutation annotation format and further processed with the R package maftools for visualization. The WES data generated in this study have been deposited in the SRA database under accession code PRJNA816435.

## PDX studies

The establishment of patient-derived xenografts has been approved by the Commission of Medical Ethics of the University Hospitals Leuven (S54185; ML8713) and by the KU Leuven EC (P038/2015) and informed consent from the patients was obtained for PDX establishment. In vivo work was done at the TRACE PDX platform

(KU Leuven), according to the ARRIVE guidelines. To establish the PDX models, tumor fragments freshly isolated from patients were implanted in the inter-scapular fat pad of female immunodeficient nude mice (NMRI-Fox1nu strain, Taconic). Mice were housed in groups of 5 in filter top cages and maintained in a semi-SPF facility at 22 °C (±2 °C) with 14 h light/10 h dark cycle and humidity between 45–70%. Bedding was replaced once a week. Mice were 5–10 weeks old at the moment of tumor implantation. Tumors were propagated in at least three generations of mice and characterized by histologic and SNP-analysis to confirm genealogy before conducting experiments on cohorts of "xenopatients".

For each treatment experiment, 25–50 mice were implanted with tumors in order to obtain enough material and statistical significance. Tumors were grown until a volume of ~300–400 mm³ was reached and were then included in treatment. Four days before actual treatment, mice were put on control diet (Baker Amino Acid Diet 5CC7, TestDiet) or serine/glycine-free diet (Mod TestDiet 5CC7 w/No Added Serine or Glycine 5BJX, TestDiet). Tumor biopsies were taken before the start of treatment. For implantation and biopsies, mice were anesthetized with 75 mg/kg ketamine (Anesketin) and 100 mg/kg medetomidine (Narcostart) intraperitoneal injection, biopsies were taken with disposable biopsy punch needles (2.5 mm, Kai Medical), postoperative care was given by 0.05 mg/kg Buprenorphine (Vetergisic) subcutaneous and reversal of anesthesia was done by Atipamezole (Antisedan) 1 mg/kg subcutaneous.

As treatment, placebo (0.9% saline) or carboplatin (50 mg/kg) was injected intraperitoneal (IP) once a week for 3 weeks. During treatment, tumor size was monitored twice a week by measuring its length ($L$) and width ($W$) with a caliper and the volume ($V$) was estimated by $V = L \times W^2 \times \pi/6$. One week after the last treatment, mice were euthanized if there was still tumor tissue present. Whenever the tumor disappeared by the end of the treatment, mice were saved and monitored for relapse. At the time of relapse, mice were treated again with carboplatin. According to the KU Leuven Ethics Committee approved protocol for our experiments (P038/2015), mice were euthanized whenever tumors reached a volume of more than 2000 mm³. This limit has been exceeded for the serine/glycine-free diet experiment with OVC2 because the model grew fast, being extremely vascularized and prone to develop liquid cysts. These data were however needed to evaluate the effect of the used diets. At any time, we monitored potential discomfort of the mice as also reported in the Ethics Committee approved document, and mice were euthanized in case any of the specified limits was reached. At euthanasia, mice were anesthetized and euthanized by cervical dislocation and blood was collected using cardiac punction. Samples were taken for protein and metabolite analysis. Proteins were extracted in Pierce RIPA Buffer (Thermo Fisher) using Lysing matrix A tubes (MP Bio) and the Precellys 24 (Bertin technologies) for lysis. Metabolites were extracted using a cryomill (Retsch) under liquid nitrogen conditions and samples were further processed as described before.

## Organoid isolation and treatment

For organoid derivation, the method described in Kopper et al. was adopted[56]. Briefly, surgical mouse tumor specimens were cut into small pieces followed by digestion with Ad-DF+++ (Advanced DMEM/F12 (Gibco) containing 1x Glutamax (Gibco), 10 mM HEPES (Gibco) and antibiotics) supplemented with 1.5 mg/ml Collagenase II (Sigma), 10 μg/ml Hyalurondiase type IV-S (Sigma), 500x primocin (InvivoGen) and 1000x Y-27632 (Rho/Rock pathway inhibitor) and incubated on an orbital shaker for 1 h at 37 °C. Suspension was strained over a 100 μm filter. After centrifugation, red blood cells were lysed in 2 ml red cell lysis buffer for 5 min at room temperature followed by additional wash steps with Ad-DF+++ medium. Next, cells were suspended in a 2:1 ratio in cold Ad-DF+++ medium:Geltrex basement membrane matrix (Gibco) and, after plating, let to solidify upside down.

Finally, ovarian cancer organoid cell growth medium with appropriate growth factors as described by Kopper et al.[56] was added. Treatment experiments were done as described by Ooft et al.[57]. Briefly, organoid cultures were dissociated using TrypLE (Gibco) for 10 min at 37 °C, filtered and replated to grow for 4 days in order to obtain uniform sized, usable organoids. Next, cultures were collected and incubated for 15 min with dispase II (Roche) to remove Geltrex, organoids were counted and seeded in clear-bottom, white-walled 96-well plates (Corning Incorporated) in 1:2 AD-DF+++:Geltrex at a concentration of 20 organoids/μl, 5 μl per well. After solidification, drugs were added in the Ad-DF+++ medium (200 μl/well). Readouts were obtained at day 0 and 5 and quantification of cell viability was done using CellTiter-Glo 3D (Promega), according to manufacturer's protocol, read using a Victor X4 Plate Reader (PerkinElmer). Pictures were taken by light microscopy. Organoids were characterized by performing H&E analysis on FFPE blocks.

### Effusion specimens and immunohistochemistry (IHC) of the Norwegian cohort

Effusions consisted of 18 HGSC specimens (16 peritoneal, 2 pleural) submitted to the Department of Pathology at the Norwegian Radium Hospital during the period of 1999 to 2008. Effusion specimens were diagnosed by an experienced cytopathologist (B.D.) based on morphology in smears and cell blocks, prepared using the thrombin clot protocol, and IHC, based on established guidelines. Informed consent was obtained according to national and institutional guidelines. Study approval was given by the Regional Committee for Medical Research Ethics in Norway (S-04300).

Formalin-fixed, paraffin-embedded sections from the 18 effusions were analyzed for PHGDH protein expression using the Dako EnVision Flex + System (K8012; Dako, Glostrup, Denmark). The PHGDH antibody was a rabbit polyclonal antibody purchased from Sigma (Prestige Antibodies, powered by Atlas Antibodies; cat # HPA021241; Stockholm, Sweden), applied at a 1:4000 dilution following antigen retrieval in Low pH buffer (pH 6.0).

Following deparaffinization and antigen retrieval, sections were treated with EnVision™ Flex + mouse linker (15 min) and EnVision™ Flex/HRP enzyme (30 min) and stained for 10 min with 3'3-diaminobenzidine tetrahydrochloride (DAB), counterstained with hematoxylin, dehydrated and mounted in Toluene-Free Mounting Medium (Dako). Positive control consisted of normal pancreas.

**IHC scoring.** Staining was scored by an experienced cytopathologist (B.D.), using a 0–4 scale for staining extent as follows: 0 = no staining, 1 = 1–5%, 2 = 6–25%, 3 = 26–75%, 4 = 76–100% of tumor cells. Staining intensity was scored as 0, 1 or 2, corresponding to negative, weak or strong staining. Values were combined by doubling, providing a 0–8 score.

The only patient with PHGDH increase at recurrence was aged between 60 and 65 years. Both mean and median age of the PHGDH equal subgroup was 50 years with SD of 12 years. The mean age of the PHGDH decreasing subgroup was 67 years, while the median age was 68 years, SD for this subgroup was 10 years.

### Tumor specimen and immunohistochemistry (IHC) of the Dutch cohort

Patients with HGSC FIGO stage IIb-IV, treated with cytoreductive surgery and adjuvant chemotherapy (PDS) or neoadjuvant chemotherapy followed by interval debulking surgery in the tertiary referral hospital The Netherlands Cancer Institute−Antoni van Leeuwenhoek Hospital (NKI-AVL) between January 2008 and December 2015 were selected. Recurrence data of the aforementioned patients who were treated in the NKI-AVL were retrieved from patient files. Last check of patient files took place on October 31, 2020. Patients from whom recurrence tissue was extracted, either via biopsy or during surgery, in the NKI-AVL were

included. For the present study, approval of the institutional review boards of the Netherlands Cancer Registry (NCR, K19.074), Dutch Pathology Registry (PALGA, 2019-169) and NKI-AVL (CFMPB297) was obtained. The requested dataset was considered anonymous and the use is therefore exempt from ethics review board approval according to Dutch legislation.

Formalin-fixed, paraffin-embedded (FFPE) tissue blocks from both the primary tumor and the recurrence samples were obtained. Diagnosis of the primary tumor was confirmed based on conventional morphological examination of sections stained with hematoxylin and eosin (H&E) by an expert pathologist (H.M.H.). The paraffin tissue blocks of the primary tumor were organized into TMAs. Representative areas for immune cell scoring of the tumor center and the peripheral invasive margin were selected on whole-tissue FFPE H&E stained slides. In each tumor four cores were selected, optimally representing tumor and peripheral stroma. TMAs with 1 mm sized cores were constructed, using a tissue microarrayer (Grand Master, Sysmex Europe GmbH, Norderstedt, Germany). To enable adhesion of the cores to the recipient paraffin block, the block was melted at 70 °C for 9 min and cooled down ON. In case of the recurrence samples, whole slides were used as samples were not large enough to obtain multiple TMA's. IHC was performed on the BenchMark Ultra autostainer (Ventana Medical Systems Inc., USA). Three μm thick TMA sections were generated and heated at 75 °C for 28 min followed by deparaffinization and rehydration. Deparaffinization was completed in the instrument using EZ prep solution (Ventana Medical Systems Inc., USA). Heat-induced antigen retrieval was initiated using Cell Conditioning 1 (Ventana Medical Systems Inc., USA) for 64 min at 95 °C. PHGDH was detected using a polyclonal antibody (cat: HPA021241) (1/8000 dilution, 64 min at 36 °C, Sigma-Aldrich). Bound antibody was detected using the OptiView DAB Detection Kit (Ventana Medical Systems). Slides were counterstained with Hematoxylin II and Bluing Reagent (Ventana Medical Systems).

**IHC scoring.** The IHC staining results of PHGDH were scored by an expert pathologist (H.M.H.) semiquantitatively by integrating intensity and distribution. Intensity was divided into categories of 0, no staining; 1, weak staining; 2, moderate staining and 3, strong staining. Staining distribution was based on the percentage of positive tumor cells (0–100%). The final PHGDH expression score was obtained by multiplying the intensity- and distribution-scores, with 0 corresponding to no staining and 300 to 100% of cells with 3+ staining intensity.

The mean and median age of the PHGDH increasing subgroup is 62 and 63 years, respectively, with a SD of 9 years. Both mean and median age of the PHGDH equal subgroup was 74 years with SD of 4 years. Both mean and median age of the PHGDH decreasing subgroup was 63 years, with a SD of 10 years.

### CCLE−TCGA data analysis

Computational data from The Cancer Genome Atlas[27] were accessed and processed using cBioPortal (http://www.cbioportal.org/)[58,59] and Microsoft Office Excel, respectively. Drug sensitivity data from the CCLE were accessed using the Genomics of Drug Sensitivity in Cancer (GDSC) database (https://www.cancerrxgene.org/)[40] and linked with gene expression data using cBioPortal.

### Data processing and statistics

Data processing was done in Microsoft Office Excel 2019; ImageJ, Microsoft Office Excel and PowerPoint 2019 was used for microscopy and western blot processing; FlowJo 10 was used for flow cytometry analysis; Matlab was used for metabolomics analysis; additional specialized softwares were used as described in the specific "Method" sections and the "Reporting summary". Figures were generated using GraphPad Prism 9. Statistical analysis was done using the available GraphPad Prism 9 tools. Each experiment was performed at least three

times as biological replicates with at least three technical replicates each time, except if stated otherwise in the figure legends. Data are represented as mean ± standard deviation or standard error. Significance was evaluated using the statistical tests and post-tests as described in the figure legends. p Values are indicated in the figures. For GI50 determination in GraphPad Prism, drug concentrations were log10 transformed and subsequently the non-linear regression built in three parameter log(inhibitor) vs. response function was used. The top constraint was put at 100, while the bottom constraint was set at 0, no constraint was set to the Hill slope.

### Reporting summary

Further information on research design is available in the Nature Research Reporting Summary linked to this article.

## Data availability

The RNA-seq data generated in this study have been deposited in the GEO database under accession code GSE176218. The WES data generated in this study have been deposited in the SRA database under accession code PRJNA816435. The used metabolomics data and the raw metabolomics data can be found in the source data file. Data from the TCGA were obtained using cBioPortal: https://www.cbioportal.org/ (Ovarian Cystadenocarincoma, TCGA, Nature 2011). Gene expression data from the CCLE were obtained using cBioPortal: https://www.cbioportal.org/ (Cancer Cell Line Encyclopedia, Broad 2019). Drug sensitivity data were obtained from the GDSC, using https://www.cancerrxgene.org/ (STUDY: olaparib, Pubchem 23725625, sample size 762, screening site Sanger, dataset GDSC2). Source data are provided with this paper.

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

## Acknowledgements

We are grateful to all past and present members of the Amant, Fendt and Agami laboratories for invaluable discussions and advice, and acknowledge the technical help of Tina Everaert and Rossana Maria Benedetto. We thank Mr. Arild Holth for the IHC staining of the effusion samples of the Norwegian cohort. We would like to acknowledge the NKI-AVL Core Facility Molecular Pathology & Biobanking (CFMPB) for supplying NKI-AVL Biobank material and lab support, for the staining of the Dutch cohort samples. We also thank TRACE, the KU Leuven PDX Platform, for technical assistance with the in vivo experiments. We would like to thank Jonas Dehairs and Frank Vanderhoydonc of the KU Leuven lipidomics core facility Lipometrix for performing lipidomics analysis. Finally, we would like to thank the Genomics Core Facilities of the Netherlands Cancer Institute and the UZ Leuven for RNA-seq analysis and WES analysis, respectively. Some schematic art pieces in Fig. 5d and Supplementary Fig. 2d were used and modified from Servier Medical Art (http://smart.servier.com/). Servier Medical Art by Servier is licensed under a Creative Commons Attribution 3.0 Unported License (https://creativecommons.org/licenses/by/3.0/). Chemical structures in Supplementary Fig. 3h are generated using PubChem Sketcher[60]. This study was supported by research grants from Kom Op Tegen Kanker—The Flemish Cancer Society (3M150511, to F.A., S.M.F. and D.L.) and KWF Kanker Bestrijding (11574, to R.A., F.A. and D.A.). F.A. is a senior researcher for Research Foundation—Flanders (FWO). T.V.N. is recipient of an Emmanuel van der Schueren fellowship from Kom op Tegen Kanker—The Flemish Cancer Society. M.R. has received consecutive postdoctoral fellowships from FWO and Stichting tegen Kanker, and J.A.G.D. was supported by FWO. S.M.F. acknowledges funding from the European Research Council under the ERC Consolidator Grant Agreement n. 771486–MetaRegulation, FWO—Research Projects (G088318N), KU Leuven FTBO, Fonds Baillet Latour, King Baudouin Foundation and the Beug Foundation. TRACE staff is supported by Stichting Tegen Kanker grant 2016-054. P.C. is supported by Grants from Methusalem funding (Flemish government), the Fund for Scientific Research-Flanders (FWO-Vlaanderen), ERC Advanced Research Grant EU-(ERC743074), and a NNF Laureate Research Grant from Novo Nordisk Foundation (Denmark).

## Author contributions

Conceptualization: T.V.N., J.A.G.D., L.V.W., H.M.H., E.A.Z., C.R.B., R.A., S.M.F., D.A. and F.A.; Methodology: T.V.N., M.P., J.A.G.D., L.V.W., H.M.H., W.D.W., D.A., M.R., E.A.Z., C.R.B., A.T., J.V.S, B.D. and S.M.F.; Investigation: T.V.N., M.P., L.V.W., J.A.G.D., R.E.M.B.A., M.R., H.M.H., S.M., A.T., G.E., E.A.Z., B.D., P.R.K. and D.A.; Resources: H.M.H., L.R., E.B., G.S.S., P.C., J.V.S., C.R.B., B.D., S.M.F., R.A., F.A.; Writing—original draft: T.V.N. and D.A.; Visualization: T.V.N.; Supervision: S.M.F., D.A. and F.A.; Funding acquisition: T.V.N., R.A., D.L., S.M.F., D.A. and F.A.

## Competing interests

S.M.F. has received funding from Bayer, Merck and BlackBelt Therapeutics and has consulted for Fund+. G.S.S. reports institutional research support from AstraZeneca, Merck, Novartis, and Roche, unrelated to this work. All the other authors declare no competing interests.

## Additional information

[1]Gynecological Oncology Laboratory, Department of Oncology, KU Leuven and Leuven Cancer Institute (LKI), 3000 Leuven, Belgium. [2]Division of Oncogenomics, Oncode Institute, The Netherlands Cancer Institute, 1066 CX Amsterdam, The Netherlands. [3]Laboratory of Cellular Metabolism and Metabolic Regulation, VIB-KU Leuven Center for Cancer Biology, VIB, Herestraat 49, 3000 Leuven, Belgium. [4]Laboratory of Cellular Metabolism and Metabolic Regulation, Department of Oncology, KU Leuven and Leuven Cancer Institute (LKI), Herestraat 49, 3000 Leuven, Belgium. [5]Department of Obstetrics and Gynecology, Maastricht University Medical Centre, Maastricht, The Netherlands. [6]GROW, School for Oncology and Developmental Biology, Maastricht, The Netherlands. [7]Department of Research, Netherlands Comprehensive Cancer Organization (IKNL), Utrecht, The Netherlands. [8]Biomolecular Mass Spectrometry and Proteomics, Bijvoet Center for Biomolecular Research, Utrecht University, 3584 CH Utrecht, The Netherlands. [9]Division of Cell Biology, Metabolism and Cancer, Department Biomolecular Health Sciences, Faculty of Veterinary Medicine, Utrecht University, 3584 CL Utrecht, The Netherlands. [10]Laboratory of Lipid Metabolism and Cancer, Department of Oncology, KU Leuven and Leuven Cancer Institute (LKI), 3000 Leuven, Belgium. [11]TRACE PDX Platform, Department of Oncology, KU Leuven and Leuven Cancer Institute (LKI), 3000 Leuven, Belgium. [12]Department of Medical Oncology, The Netherlands Cancer Institute, 1066 CX Amsterdam, The Netherlands. [13]Division of Molecular Pathology, The Netherlands Cancer Institute, 1066 CX Amsterdam, The Netherlands. [14]Laboratory of Angiogenesis and Vascular Metabolism, VIB-KU Leuven Center for Cancer Biology, VIB, Department of Oncology, KU Leuven and Leuven Cancer Institute (LKI), 3000 Leuven, Belgium. [15]Department of Development and Regeneration, KU Leuven, 3000 Leuven, Belgium. [16]Center for Biotechnology, Khalifa University of Science and Technology, Abu Dhabi, United Arab Emirates. [17]Laboratory of Angiogenesis and Vascular Heterogeneity, Department of Biomedicine, Aarhus University, 8000 Aarhus, Denmark. [18]Laboratory for Translational Genetics, Department of Human Genetics, KU Leuven, 3000 Leuven, Belgium. [19] VIB Center for Cancer Biology, VIB, 3000 Leuven, Belgium. [20]University of Oslo, Faculty of Medicine, Institute of Clinical Medicine, N-0316 Oslo, Norway. [21]Department of Pathology, Oslo University Hospital, Norwegian Radium Hospital, N-0310 Oslo, Norway. [22]Erasmus MC, Department of Genetics, Rotterdam University, 3015 GD Rotterdam, The Netherlands. [23]Department of Obstetrics and Gynecology, University Hospitals Leuven and Department of Oncology, 3000 Leuven, Belgium. [24]Centre for Gynecologic Oncology Amsterdam (CGOA), Antoni Van Leeuwenhoek-Netherlands Cancer Institute (AvL-NKI), University Medical Center (UMC), Amsterdam, The Netherlands. [25]These authors contributed equally: Daniela Annibali, Frédéric Amant. ✉e-mail: daniela.annibali@kuleuven.be; frederic.amant@uzleuven.be

