## [Peer Review File · Nature Communications]

Serine metabolism remodeling after platinum-based chemotherapy identifies vulnerabilities in a subgroup of resistant ovarian cancersThis manuscript has been previously reviewed at another journal that is not operating a transparent peer review scheme. This document only contains reviewer comments and rebuttal letters for versions considered at Nature Communications.

REVIEWERS' COMMENTS

Reviewer #1 (Remarks to the Author):

The authors have made substantial edits to the manuscript and included a large amount of new data that respond appropriately and thoroughly to my queries. The manuscript is suitable for publication and I am looking forward to seeing it in print.

Reviewer #2 (Remarks to the Author):

The authors have addressed all my concerns as best as can be done eg with the availability of HRD assessments at this time. I believe that it is more useful to have what information about BRCA-like mutations and from WES is available, in the manuscript than not, and to refer to the issue of HRDness in regards to PARPi response, because leaving it out altogether would be confusing for the field, especially as PARPi progressively occupy an even larger role in the first-line context going forward. It is also appropriate to have included all the characterisation of the PDX which looks very good.

Thank you for including the data from SF3 (k-1) in main Fig 5 - the possibility of using NAD+ synthesis inhibitors in combination with PARPi is now shown in the main manuscript to have some merit worthy of future consideration and will be easier for all to see. It is required for the logical extension of the argument in the manuscript.

Having cultured many cell types in PARPi drug assays, maximal killing is often observed from 7-14 days eg at 10 days, so ensure you are optimising your culture conditions in a suitable set up, with replenishment of media/drugs etc. PARPi-mediated death can take time as your assays show - you may not have seen the optimal combinatorial death - I cant tell from your graphs.

We would like to thank both reviewers for the time and attention they devoted to improve our manuscript, and we are glad they appreciated our efforts to address all their previous comments.

Reviewer #1 (Remarks to the Author):

The authors have made substantial edits to the manuscript and included a large amount of new data that respond appropriately and thoroughly to my queries. The manuscript is suitable for publication and I am looking forward to seeing it in print.

We believe that the quality of the work has substantially increased thanks to the reviewer's original comments and constructive observations, and we are glad we could address them all.

Reviewer #2 (Remarks to the Author):

The authors have addressed all my concerns as best as can be done eg with the availability of HRD assessments at this time. I believe that it is more useful to have what information about BRCA-like mutations and from WES is available, in the manuscript than not, and to refer to the issue of HRDness in regards to PARPi response, because leaving it out altogether would be confusing for the field, especially as PARPi progressively occupy an even larger role in the first-line context going forward. It is also appropriate to have included all the characterisation of the PDX which looks very good.

We agree with reviewer #2 that HRD status is of critical importance in this research field and we are thankful that he/she appreciates the WES analysis we did on all our research models in order to address this concern as good as possible.

Thank you for including the data from SF3 (k-1) in main Fig 5 - the possibility of using NAD⁺ synthesis inhibitors in combination with PARPi is now shown in the main manuscript to have some merit worthy of future consideration and will be easier for all to see. It is required for the logical extension of the argument in the manuscript.

The changes in the figures make the results more consistent, and provide a better rationale for future research. We are happy that the reviewer appreciates this.

Having cultured many cell types in PARPi drug assays, maximal killing is often observed from 7-14 days eg at 10 days, so ensure you are optimising your culture conditions in a suitable set up, with replenishment of media/drugs etc. PARPi-mediated death can take time as your assays show - you may not have seen the optimal combinatorial death - I cant tell from your graphs.

The possibility to extend the combination treatment is a very interesting suggestion. We originally chose to treat our cell lines for 5 days, to evaluate the results when most of the experiments were performed and the control cells reached confluency. Although the effect of the combination treatment is clearly present after 5 days, the possibility that the combinatorial effect might be even stronger after 7-14 days is compelling. We repeated the combination treatment in the A2780cis cells for 7 days, and we observed this time that after 5-7 days the cells started to die in the combination treatment arm, while this was not the case in the single treatment arm (Reviewer’s figure 1). However, we were only able to do this experiment n=1 time with n=3 technical replicates due to limited available time.

We believe that, for future research, additional optimizations in doses and treatment schemes will be necessary in order to obtain the maximal response, and that this will also depend on the cell lines and/or research models that are used. This is also the reason why, in line with reviewer’s comment, in our organoid lines we observed a clear effect of the combination treatment only after 7-10 days (Manuscript Figure 7).

Reviewer’s figure 1: carboplatin + olaparib treatment in A2780cis cells for 7 days. (A) Growth of cis cells under carboplatin/olaparib treatment and its combination. (B) Cell death of cis cells under carboplatin/olaparib treatment and its combination. N=1 biological replicate with n=3 technical replicates.